# Cells use molecular working memory to navigate in changing chemoattractant fields

Akhilesh Nandan[†‡], Abhishek Das[†‡], Robert Lott[‡], Aneta Koseska*[‡]

Department of Systemic Cell Biology, Max Planck Institute of Molecular Physiology, Dortmund, Germany

**Abstract** In order to migrate over large distances, cells within tissues and organisms rely on sensing local gradient cues which are irregular, conflicting, and changing over time and space. The mechanism how they generate persistent directional migration when signals are disrupted, while still remaining adaptive to signal's localization changes remain unknown. Here, we find that single cells utilize a molecular mechanism akin to a working memory to satisfy these two opposing demands. We derive theoretically that this is characteristic for receptor networks maintained away from steady states. Time-resolved live-cell imaging of Epidermal growth factor receptor (EGFR) phosphorylation dynamics shows that cells transiently memorize position of encountered signals via slow-escaping remnant of the polarized signaling state, a dynamical 'ghost', driving memory-guided persistent directional migration. The metastability of this state further enables migrational adaptation when encountering new signals. We thus identify basic mechanism of real-time computations underlying cellular navigation in changing chemoattractant fields.

**\*For correspondence:**
aneta.koseska@mpinb.mpg.de

[†]These authors contributed equally to this work

**Present address:** [‡]Cellular Computations and Learning, Max Planck Institute for Neurobiology of Behavior - caesar, Bonn, Germany

**Competing interest:** The authors declare that no competing interests exist.

## Editor's evaluation

This paper addresses how cells can robustly maintain direction during movement by ignoring noise in concentration gradients while also being able to adapt to new signals in those gradients. The authors study this tension in EGFR signaling by postulating a form of cellular memory in a theoretical framework based on dynamical systems and bifurcation theory. The authors also carry out experiments that raise further interesting questions. This paper will be of interest to scientists of all stripes working on cell motility and for theorists who take a dynamical systems view of biological phenomena.

## Introduction

Directed chemotactic behavior relies on generating polarized signaling activity at the plasma membrane of the cell that is translated to an elongated cell shape, and subsequent persistent migration in the direction of the signal. Experimental observations have shown that cells as diverse as social amoeba, neutrophils, leukocytes, fibroblasts, and nerve cells maintain the acquired orientation even when signals are disrupted or noisy (*Parent and Devreotes, 1999*; *Foxman et al., 1999*; *Ridley et al., 2003*). However, not only do they respond robustly to dynamic gradients, they can also adapt the migrational direction by integrating and resolving competing spatial signals, or prioritizing newly encountering attractants (*Jilkine and Edelstein-Keshet, 2011*; *Skoge et al., 2014*; *Albrecht and Petty, 1998*). This suggests that cells likely memorize their recent environment. Numerous models based on positive feedbacks, incoherent feed-forward, excitable or Turing-like networks have been proposed to describe how polarized signaling activity of cell-surface receptors and/or downstream

**eLife digest** If we are injured, or fighting an infection, cells in our body migrate over large distances to the site of the wound or infection to act against any invading microbes or repair the damage. Cells navigate to the damaged site by sensing local chemical cues, which are irregular, conflicting and change over time and space. This implies that cells can choose which direction to travel, and stick to it even if the signals around them are disrupted, while still retaining the ability to alter their direction if the location of the signal changes. However, how cells are able to effectively navigate their way through this field of complex chemical cues is poorly understood.

To help resolve this mystery, Nandan, Das et al. studied the epidermal growth factor receptor (EGFR) signaling network which controls how some cells in the body change shape and migrate. The network is activated by specific chemical cues, or ligands, binding to EGFR proteins on the cell surface. The receptors then join together to form pairs, and several tags known as phosphate groups are added to each molecule. This process (known as phosphorylation) switches the receptor pair to an active state, allowing EGFR to relay signals to other proteins in the cell and promote the activity of receptors not bound to a ligand. The phosphorylation state of EGFRs is then modulated over time and across the cell by a network of enzymes called phosphatases which can remove the phosphate groups and switch off the receptor.

To study EGFR phosphorylation dynamics in human cells, Nandan, Das et al. imaged individual cells over time using a microscope. This data was then combined with a mathematical model describing the EGFR signaling network and how cells change their shape over time.

The experiment revealed that the phosphate groups attached to EGFR are not removed immediately when the chemical cue is gone. Instead, the active state is transiently maintained before complete inactivation. This had the effect of encoding a short-lived memory in the signaling network that allowed the cells to continue to migrate in a certain direction even when chemical cues were disrupted. This memory state is dynamic, enabling cells to adapt direction when the cue changes location.

The findings of Nandan, Das et al. reveal the underlying mechanism for how cells decipher complex chemical cues to migrate to where they are needed most. The next steps to follow on from this work will be to understand if other receptors involved in migration work in a similar way.

signaling component such as members of the Rho GTPase family can arise (*Levchenko and Iglesias, 2002*; *Levine et al., 2002*; *Mori et al., 2008*; *Goryachev and Pokhilko, 2008*; *Beta et al., 2008*; *Xiong et al., 2010*; *Trong et al., 2014*; *Halatek and Frey, 2018*). This polarized activity in turn controls actin and myosin dynamics, and thereby cell migration. Conceptually, the underlying dynamical principles of the proposed models are similar, and can be understood as switching from the stable state of basal- to the stable polarized-signaling steady state in presence of guiding external cues. However, they can account either for sensing and adaptation to non-stationary stimuli or for long-term maintenance of polarized signaling activity, but not both. Thus, how cells process the information from a changing chemoattractant field in real time for long-range navigation remains unknown.

We propose a shift in the conceptual framework, describing theoretically that efficient navigation can be achieved when the polarized signaling state of the receptor network is transiently stable. This is fulfilled in the presence of dynamical 'ghosts' at a unique dynamical transition, which we demonstrate in the EGFR signaling network dynamics using a mathematical model, as well as quantitative live-cell imaging of polarized EGFR signaling. We show with a physical model of the cell and migration experiments using microfluidics, that cells generate memory of encountered signals through the 'ghost' state, translating it to memory in polarized shape changes and directional migration. Due to the metastability of the 'ghost' state, cells can also easily adapt their migration direction depending on the changes in signal localization. We therefore describe a basic mechanism of real-time cellular navigation in complex chemoattractant fields.

## Results

### Dynamical mechanism of navigation in non-stationary environments

We conjectured that only dynamically metastable receptor signaling states can enable both transient stability of polarized signaling as necessary for robust, memory-guided migration in noisy fields, as well as rapid adaptation of its direction when signals vary in space and time. Our hypothesis is that this can be achieved if biochemical systems are maintained outside, but in the vicinity of the polarization steady state. We therefore approached the problem using the abstract language of dynamical systems theory, where the characteristics of any process directly follow from the type of dynamical transitions, called bifurcations, through which they emerge (*Strogatz, 2018*).

Directed migration relies on a polarized representation of the directional signal, requiring a reliable mechanism for signal-induced transition from a non-polarized symmetric, to a polarized receptor signaling state, and subsequently polarized cell shape. This transition is thus a symmetry-breaking transition, and we propose that a pitchfork bifurcation (*PB*, *Koseska et al., 2013*; *Strogatz, 2018*) satisfies the necessary dynamical conditions (*Figure 1A*, *Figure 1—figure supplement 1A*). Transient memory on the other hand is a unique characteristic of another bifurcation, a saddle-node (*SN*) bifurcation, that characterizes a transition between stable and unstable steady states. When the *SN* and thereby a stable steady-state is lost, for example upon signal removal, a remnant or a dynamical 'ghost' of the stable state emerges (*Strogatz, 2018*). These 'ghost' states are dynamically metastable and transiently maintain the system in the vicinity of the steady state (*Figure 1A*, *Figure 1—figure supplement 1A*). Necessary for manifestation of the 'ghost' state is organization at criticality, before the *SN*. We have previously examined both theoretically and experimentally, the response of receptor networks under uniform growth factor stimulation and determined that the concentration of receptors on the cell membrane regulate the organization of the system at criticality (*Stanoev et al., 2018*; *Stanoev et al., 2020*). The features of both bifurcations, cell polarization under spatial cues and a transient memory of this polarization in absence of the cue, will be unified for a sub-critical *PB*, as it is stabilized via $SN_{PB}$s. We thus propose that organization at criticality - in the vicinity of a $SN_{PB}$ (gray shaded area in *Figure 1—figure supplement 1A*; details discussed in Materials and methods), renders a minimal mechanism for cellular responsiveness in changing environments.

We described this conjecture mathematically for a general reaction-diffusion model representing the signaling activity on the plasma membrane of a cell, $\frac{\partial \mathbf{U(x,t)}}{\partial t} = \mathbf{F(U)} + \mathbf{D}\nabla^2\mathbf{U(x,t)}$, with $\mathbf{U}$ being the vector of local densities of active signaling components, $\mathbf{D}$ - diffusion constants and $\mathbf{F}$ accounting for all chemical reactions. Our theoretical analysis shows that a *PB* exists if, for a spatial perturbation of the symmetric steady state ($\mathbf{U_s}$) of the form $\mathbf{U(x},t) = \mathbf{U_s} + \delta\mathbf{U(x)}e^{\lambda t}$, the conditions $\delta\mathbf{U(-x)} = -\delta\mathbf{U(x)}$ and the limit $\lim_{\lambda \to 0} F_\lambda = det(J) = 0$ are simultaneously fulfilled (Materials and methods). This implies that the linearized system has zero-crossing eigenvalues ($\lambda$) associated with the odd mode of the perturbation (*Paquin-Lefebvre et al., 2020*). To probe the sub-critical transition and therefore the necessary organization at criticality, a reduced description in terms of an asymptotic expansion of the amplitude of the polarized state ($\phi$) must yield the Landau equation $\frac{d\phi}{dt} = c_1\phi + c_2\phi^3 - c_3\phi^5$, guaranteeing the existence of $SN_{PB}$ (see Materials and methods for derivation).

These abstract dynamical transitions can be realized in receptor signaling networks with different topologies and are best analyzed using computational models, whose predictions are then tested in quantitative experiments on living cells. To exemplify the above-mentioned principle, we use the well-characterized Epidermal growth factor receptor (EGFR) sensing network (*Reynolds et al., 2003*; *Baumdick et al., 2015*; *Stanoev et al., 2018*). It constitutes of double negative and negative feedback interactions of the receptor, EGFR ($E_p$) with two enzymes, the phosphatases PTPRG ($P_{RG}$) and PTPN2 ($P_{N2}$, *Figure 1B*, *Figure 1—figure supplement 1B*), respectively. $E_p$ and $P_{RG}$ laterally diffuse on the membrane and inhibit each-other's activities (see Materials and methods for the molecular details of the network). The bidirectional molecular interactions between EGFR and the phosphatases can be mathematically represented using mass action kinetics, giving a system of partial differential equations (PDE) that describes how the dynamics of the constituents evolves in time and space (*Equation 14* in Materials and methods). Applying a weakly nonlinear stability analysis (*Becherer et al., 2009*) to this system of equations shows that the EGFR phosphorylation dynamics undergoes a symmetry-breaking transition (*PB*) as outlined above (proof in Materials and methods, *Figure 1—figure supplement 1C*). The *PB* generates a polarized state that is represented as a inhomogeneous steady state (IHSS) - a combination of a high receptor phosphorylation at the cell front and low in the back of the cell

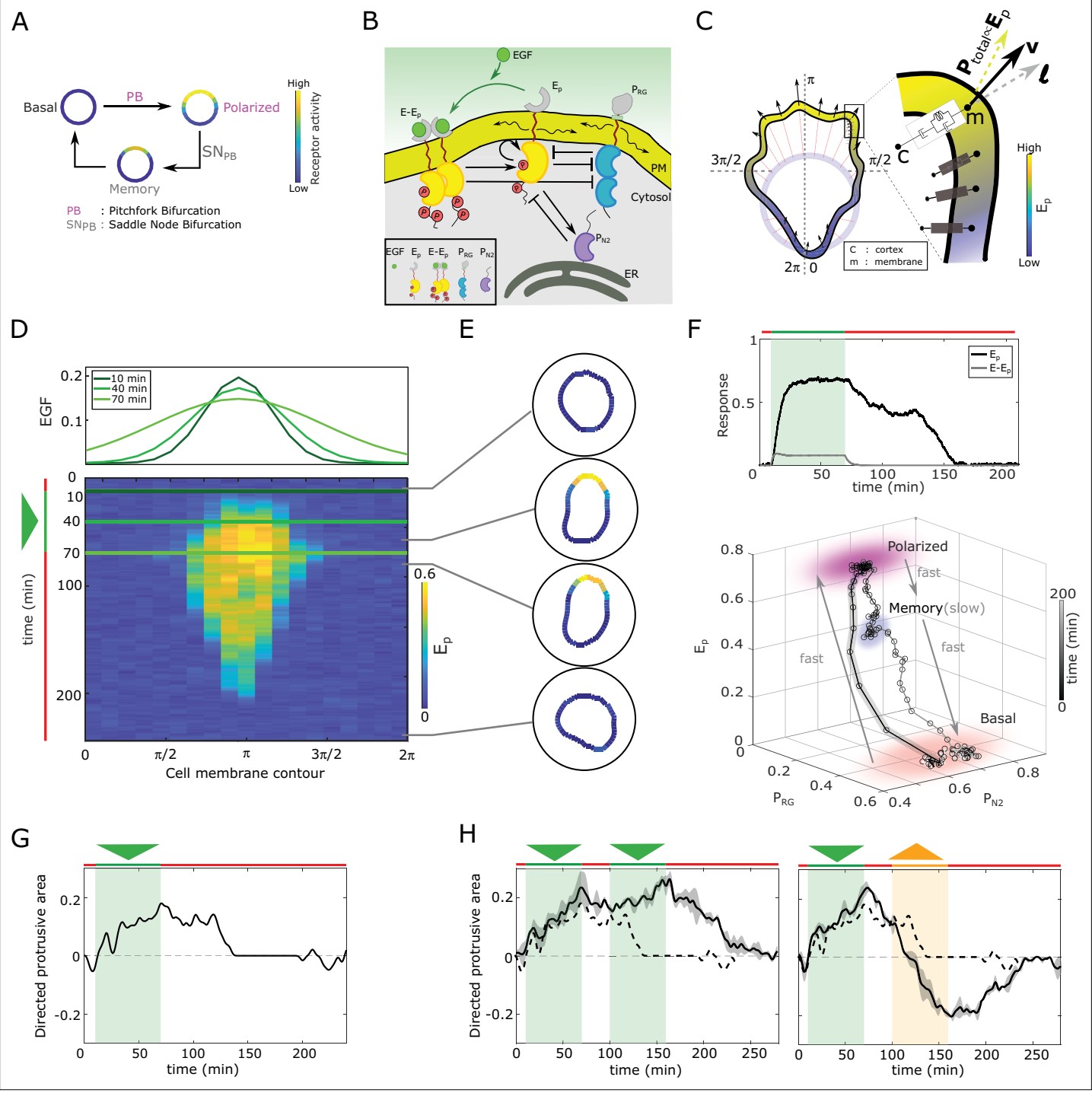

**Figure 1.** In silico manifestation of metastable polarized membrane signaling, as a mechanism for sensing changing spatial-temporal signals. (**A**) Dynamical mechanism: sub-critical pitchfork bifurcation ($PB$) determines stimulus-induced transition (arrow) between basal unpolarized and polarized receptor signaling state, whereas the associated saddle-node through which the $PB$ is stabilized ($SN_{PB}$) gives rise to a 'ghost' memory state upon signal removal for organization at criticality (before the $SN_{PB}$). See *Figure 1—figure supplement 1A* and Methods for detailed description of these transitions. (**B**) Scheme of the EGFR-PTP interaction network. Ligandless EGFR ($E_p$) interacts with PTPRG ($P_{RG}$) and PTPN2 ($P_{N2}$). Liganded EGFR ($E - E_p$) promotes autocatalysis of $E_p$. Causal links - solid black lines; curved arrow lines - diffusion, PM - plasma membrane, ER- endoplasmic reticulum. See also *Figure 1—figure supplement 1B*. (**C**) Signal-induced shape-changes during cell polarization. Arrows: local edge velocity direction. Zoom: Viscoelastic model of the cell - parallel connection of an elastic and a viscous element. $\mathbf{P_{total}}$: total pressure; $\mathbf{v}$: local membrane velocity; l: viscoelastic state. Bold letters: vectors. Cell membrane contour: $[0, 2\pi]$. (**D**) Top: In silico evolution of spatial EGF distribution. Bottom: Kymograph of $E_p$ for organization at criticality from reaction-diffusion simulations of the network in (**B**). Triangle - gradient duration. (**E**) Corresponding exemplary

*Figure 1 continued on next page*

*Figure 1 continued*

cell shapes with color coded $E_p$, obtained with the model in (**C**). (**F**) Top: Temporal profiles $E_p$ (black) and $E - E_p$ (gray). Green shaded area: EGF gradient presence. Bottom: State-space trajectory of the system with denoted trapping state-space areas (colored) and respective time-scales. See also *Figure 1—video 1*. Thick/thin line: signal presence/absence. (**G**) Quantification of in silico cell morphological changes from the example in (**E**). Triangle - gradient duration. (**H**) Left: same as in (**G**), only when stimulated with two consecutive dynamic gradients (triangles) from same direction. Second gradient within the memory phase of the first. See also *Figure 1—figure supplement 1D*. Right: the second gradient (orange triangle) has opposite direction. See also *Figure 1—figure supplement 1E*. Dashed line: curve from (**G**). Mean ± s.d. from n=3 is shown. Parameters: Materials and methods. In (**D-H**), green(orange)/red lines: stimulus presence/absence.

The online version of this article includes the following video and figure supplement(s) for figure 1:

**Figure supplement 1.** Features of receptor activity for different organization in parameter space.

**Figure 1—video 1.** Corresponding to Figure 1F.

https://elifesciences.org/articles/76825/figures#fig1video1

(schematically shown in *Figure 1A*, *Figure 1—figure supplement 1A*). This is contrary to a bistable system, where the polarized signaling state would be manifested by two steady states, high and low protein phosphorylation in the front and back of the cell, respectively (*Beta et al., 2008*). This profiles *PB* as a robust mechanism of cell polarization. Polarized EGFR signaling on the other hand, will lead to reorganization of the cortical actomyosin cytoskeleton by regulating members of the Rho GTPase family, thereby inducing signal-dependent cell shape changes and subsequent migration (*Chiasson-MacKenzie and McClatchey, 2018*; *Ridley and Hall, 1992*). In order to link signaling activity with morphodynamics, we modeled the cell as a viscoelastic cortex surrounding a viscous core (*Yang et al., 2008*) (Materials and methods), where EGFR signaling dynamics affects cell shape changes through the protrusion/retraction stress and the viscoelastic nature of the cell membrane (*Figure 1C*).

We first fixed the total EGFR concentration on the cell membrane to a value that corresponds to organization at criticality, and investigated the response of the in silico cell to gradient stimulus. In the absence of stimulus, basal EGFR phosphorylation is uniformly distributed along the cell membrane rendering a symmetrical cell shape (*Figure 1D and E*). Introducing dynamic gradient stimulus in the simulation (slope changes from steep to shallow over time, *Figure 1D*, top) led to rapid polarization of EGFR phosphorylation in the direction of the maximal chemoattractant concentration, generating a cell shape with a clear front and back. The polarized signaling state was maintained for a transient period of time after removal of the gradient, corresponding to manifestation of memory of the localization of the previously encountered signal (*Figure 1D and E*; temporal profile *Figure 1F*, top). The prolonged polarized state does not result from remnant ligand-bound receptors ($E - E_p$) on the plasma membrane, as they exponentially decline after signal removal (*Figure 1F*, top). The memory in polarized signaling was also reflected on the level of the cell morphology, as shown by the difference of normalized cell protrusion area in the front and the back of the cell over time (*Figure 1G*). Plotting the trajectory that describes the change of the state of the system over time (state-space trajectory, *Figure 1F* bottom) shows that the temporal memory in EGFR phosphorylation polarization is established due to transient trapping of the signaling state trajectory in state-space, a property of the metastable 'ghost' state (*Stanoev et al., 2020*; *Strogatz, 2018*) through which the system is maintained away from the steady state. The simulations show that there are two characteristic time-scales present in the system: slow evolution of the system's dynamics in the 'ghost' state due to the trapping, and fast transitions between the steady states (*Figure 1—video 1*). This emergence of the slow time-scale is another hallmark of systems organized at criticality. What is crucial here however, is that the trapping in the dynamically metastable memory state does not hinder sensing of, and adapting to subsequent signals. The cell polarity is sustained even when the EGF signal is briefly disrupted (*Figure 1H* left, *Figure 1—figure supplement 1D*), but also, the cell is able to rapidly reverse direction of polarization when the signal direction is inverted (*Figure 1H* right, *Figure 1—figure supplement 1E*).

We next chose in the simulations a higher EGFR concentration on the membrane, such that the system moves from criticality to organization in the stable polarization state (magenta lines, *Figure 1—figure supplement 1C*). In this scenario, even a transient signal induces switching to the polarized state that is permanently maintained, generating a long-term memory of the direction on the initial signal. Thus, the cell is insensitive to subsequent stimuli from the same direction, whereas consecutive gradients from opposite directions generate conflicting information that cannot be resolved (*Figure 1—figure supplement 1F*). Organization in the homogeneous, symmetric steady states on

the other hand renders cells insensitive to the extracellular signals (*Figure 1—figure supplement 1G,H*). These response features for organization in the stable steady state regimes resemble the finding of the previously published models: such models cannot simultaneously capture memory in polarization along with continuous adaptation to novel signals, or require fine-tuning of kinetic parameters to explain the experimentally observed cell behavior (*Levchenko and Iglesias, 2002*; *Levine et al., 2002*; *Mori et al., 2008*; *Goryachev and Pokhilko, 2008*; *Beta et al., 2008*; *Xiong et al., 2010*; *Trong et al., 2014*). This demonstrates that organization at criticality, in a vicinity of a $SN_{PB}$, is a unique mechanism for processing changing signals.

## Cells display temporal memory in polarized receptor phosphorylation resulting from a dynamical 'ghost'

To test experimentally whether cells maintain memory of the direction of previously encountered signals through prolonged EGFR phosphorylation polarization, and what is the duration of this effect, epithelial breast cancer-derived MCF7 cells were subjected for 1 hr to a stable gradient of fluorescently tagged EGF-Alexa647 (EGF$^{647}$) with a maximal amplitude of 10 ng/ml applied from the top of the chamber in a computer-programmable microfluidic device (*Figure 2A and B*). EGFR phosphorylation at the plasma membrane was quantified during and for 3 hr after gradient wash-out (gradient wash-out established in 4–5 min) by determining the rapid translocation of mCherry-tagged phosphotyrosine-binding domain (PTB$^{mCherry}$) to phosphorylated tyrosines 1086/1148 of ectopically expressed EGFR-mCitrine (EGFR$^{mCitrine}$) using ratiometric imaging (*Offterdinger et al., 2004*; Materials and methods). Due to the low endogenous EGFR levels in MCF7 cells, the expression range of EGFR$^{mCitrine}$ was set to mimic the endogenous receptor range in the related MCF10A cell line, such that both cell lines have equivalent signaling properties of downstream effector molecules (*Stanoev et al., 2018*), and were therefore used in a complementary way in this study.

Kymograph analysis of $EGFR^{mCitrine}$ phosphorylation at the plasma membrane of single cells showed polarization in a shallow gradient of EGF$^{647}$ (as shallow as 10% between front and back of the cell; *Figure 2C*, *Figure 2—figure supplement 1A-D*). The direction of $EGFR^{mCitrine}$ phosphorylation polarization coincided with the direction of maximal EGF$^{647}$ concentration around each cell ($\pi/4$ on average, *Figure 2—figure supplement 1F*). Only few cells manifested basal or symmetric $EGFR^{mCitrine}$ phosphorylation distribution upon gradient stimulation (*Figure 2—figure supplement 1A, B, E*). Plotting the fraction of plasma membrane area with polarized $EGFR^{mCitrine}$ phosphorylation showed cell-to-cell variability in the polarization kinetics, as well as the maximal amplitude of polarized $EGFR^{mCitrine}$ phosphorylation (*Figure 2—figure supplement 1G*), in contrast to the rapid EGFR polarization in the numerical simulations (*Figure 1D*). These differences likely results from the variable positioning of the cells along the gradient in the microfluidic chamber, as well as the variability of total EGFR concentrations in single cells. However, quantification of the polarization duration revealed that, similarly to the numerical predictions, the polarization persisted ~ 40 min on average after gradient removal ([4–159 min], *Figure 2D and E*).

The memory in $EGFR^{mCitirne}$ phosphorylation was also reflected in the respective single-cell temporal profiles (exemplary profile shown in *Figure 2F*, top). Reconstructing the state-space trajectory from this temporal profile using Takens's delay embedding theorem (*Takens, 1980*) (Materials and methods) showed that before the fast transition to the basal state, the trajectory of the system was trapped in the vicinity of the polarized state (*Figure 2F* bottom, *Figure 2—video 1*). Despite the biological and technical noise that affect the measurement of the temporal $EGFR^{mCitrine}$ phosphorylation profile, and thereby the reconstruction of the state-space trajectory, both qualitatively resemble the equivalent numerical profiles (compare *Figures 1F and 2F*). In contrast, when cells were subjected to an ATP analog EGFR inhibitor (*Björkelund et al., 2013*) during gradient wash-out, the $EGFR^{mCitrine}$ phosphorylation response exponentially decayed, resulting in a clear absence of transient memory and respective state-space trapping (*Figure 2G*, *Figure 2—figure supplement 2A, B*, *Figure 2—video 2*). Since Lapatinib inhibits the kinase activity of the receptor, the dynamics of the system in this case is mainly guided by the dephosphorylating activity of the phosphatases. Implementing an equivalent of the Lapatinib inhibition in the numerical simulations by decreasing the autocatalytic EGFR activation rate constant after gradient removal verifies that the presence of memory in EGFR phosphorylation cannot be explained only by a dephosphorylation process (*Figure 2—figure supplement 2C*). This is also evident from the respective state-space trajectory, where the system directly transits

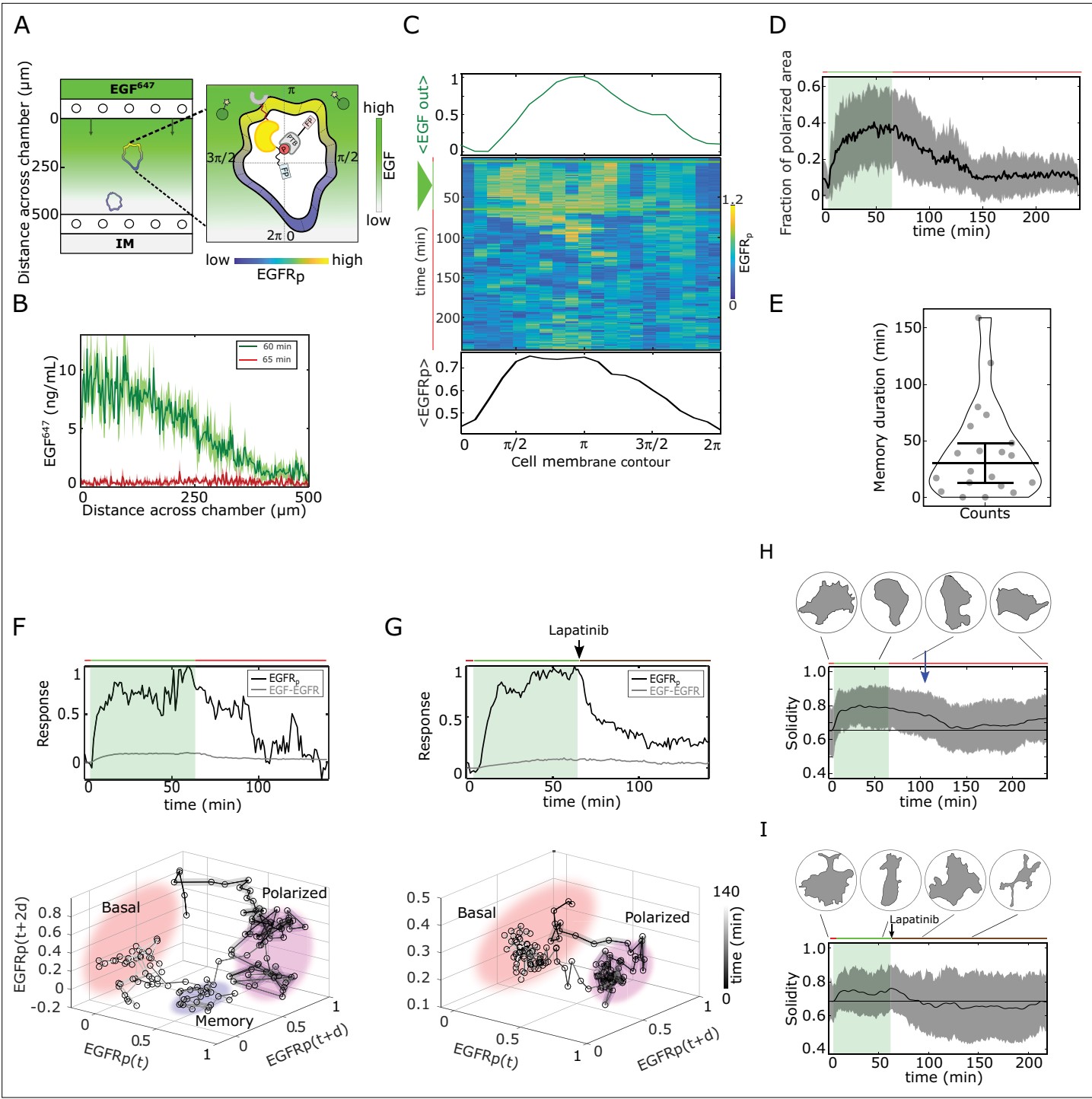

**Figure 2.** Molecular memory in polarized $EGFR^{mCitrine}$ phosphorylation resulting from dynamical state-space trapping is translated to memory in polarized cell shape. (**A**) Scheme of microfluidic $EGF^{647}$-gradient experiment; Zoom: single-cell measurables. Cell membrane contour $[0, 2\pi]$ (20 segments). $PTB$ - phosphotyrosine binding domain, $FP$/star symbol - fluorescent protein, $EGFR_p$- phosphorylated $EGFR^{mCitrine}$. Remaining symbols as in *Figure 1B*. (**B**) Quantification of $EGF^{647}$ gradient profile (at $60min$, green) and after gradient wash-out (at $65min$, red). Mean ± s.d., N=4. (**C**) Exemplary quantification of, Top: Spatial projection of $EGF^{647}$ around the cell perimeter. Gaussian fit of the spatial projection is shown. Middle: single-cell $EGFR_p$ kymograph. Data was acquired at 1 min intervals in live $MCF7 - EGFR^{mCitrine}$ cells subjected for $60min$ to an $EGF^{647}$ gradient. Other examples in *Figure 2—figure supplement 1D*. Bottom: respective spatial projection of $EGFR_p$. Gaussian fit of the spatial projection is shown. Mean ± s.d. from n=20 cells, N=7 experiments in *Figure 2—figure supplement 1C*. (**D**) Average fraction of polarized plasma membrane area (mean ± s.d.). Single cell profiles in *Figure 2—figure supplement 1G*. (**E**) Quantification of memory duration in single cells (median±C.I.). In (**D**) and (**E**), n=20, N=7. (**F**) Top: Exemplary temporal profiles of phosphorylated $EGFR^{mCitrine}$ (black) and $EGF^{647} - EGFR^{mCitrine}$ (gray) corresponding to (**C**). Bottom: Corresponding

*Figure 2 continued on next page*

*Figure 2 continued*

reconstructed state-space trajectory (*Figure 2—video 1*) with denoted trapping state-space areas (colored). Thick/thin line: signal presence/absence. d - embedding time delay. (**G**) Equivalent as in (**F**), only in live MCF7-EGFR $SN_{PB}$ cell subjected to 1 hr EGF$^{647}$ gradient (green shading), and 3 hr after wash-out with 1 μM Lapatinib. Corresponding kymograph shown in *Figure 2—figure supplement 2A*. Mean ± s.d. temporal profile from n=9, N=2 in *Figure 2—figure supplement 2B*. Bottom: Corresponding reconstructed state-space trajectory with state-space trapping (colored) (Methods, *Figure 2—video 2*). (**H**) Averaged single-cell morphological changes (solidity, mean ± s.d. from n=20, N=7). Average identified memory duration (blue arrow): 40*min*. Top insets: representative cell masks at distinct time points. (**I**) Average solidity in $MCF7 - EGFR^{mCitrine}$ cells subjected to experimental conditions as in (**G**). Mean ± s.d. from n=9, N=2. Top insets: representative cell masks at distinct time points. In (**F-I**) green shaded area: EGF$^{647}$ gradient duration; green/red lines: stimulus presence/absence. Brown line: Lapatinib stimulation. See also *Figure 2—figure supplements 1 and 2*.

The online version of this article includes the following video, source data, and figure supplement(s) for figure 2:

**Source data 1.** Source data for *Figure 2*.

**Figure supplement 1.** Quantification of $EGFR^{mCitrine}$ phosphorylation polarization.

**Figure supplement 1—source data 1.** Source data for *Figure 2—figure supplement 1*.

**Figure supplement 2.** Memory in polarized $EGFR_p$ results from a dynamical 'ghost'.

**Figure supplement 2—source data 1.** Source data for *Figure 2—figure supplement 2*.

**Figure 2—video 1.** Corresponding to Figure 2F.
https://elifesciences.org/articles/76825/figures#fig2video1

**Figure 2—video 2.** Corresponding to Figure 2G.
https://elifesciences.org/articles/76825/figures#fig2video2

**Figure 2—video 3.** Corresponding to Figure 2—figure supplement 2D.
https://elifesciences.org/articles/76825/figures#fig2video3

from the polarized to the basal state, without intermediate state-space trapping (*Figure 2—figure supplement 2D*, *Figure 2—video 3*).

Fitting the experimentally measured single-cell temporal $EGFR^{mCitrine}$ phosphorylation profiles after gradient wash-out using an inverse sigmoid function (Methods) further corroborated that under Lapatinib treatment, phosphorylated $EGFR^{mCitrine}$ exponentially relaxed from the polarized to the basal state (Hill coefficient ≈ 1.28), with a half-life of approx. 10 min (*Figure 2—figure supplement 2E, G*). Under normal conditions however, the half-life was 30 min on average, reflecting that the phosphorylated $EGFR^{mCitrine}$ is transiently maintained in the metastable signaling state after gradient removal, before rapidly switching to the basal state (Hill coefficient ≈2.88, *Figure 2—figure supplement 2F, G*). Taken together, this analysis suggests that the memory in polarized $EGFR^{mCitrine}$ phosphorylation results from a dynamically metastable 'ghost' state, and not a slow dephosphorylation process.

In order to identify whether the memory in polarized $EGFR^{mCitrine}$ phosphorylation also enables maintaining memory of polarized cell morphology after gradient removal, we quantified the cellular morphological changes using solidity, which is the ratio between the cell's area and the area of the convex hull. The average single-cell solidity profile over time showed that epithelial cells maintained the polarized cell shape for ~ 40 min after signal removal (*Figure 2H*, Materials and methods), which directly corresponds to the average memory duration in polarized $EGFR^{mCitrine}$ phosphorylation (*Figure 2E*). The exemplary quantification of the temporal evolution of the cell protrusion area in direction of the gradient showed equivalent results (*Figure 2—figure supplement 2H* corresponding to the profile in *Figure 2C*; memory duration ~ 43 min). In contrast, the absence of memory in $EGFR^{mCitrine}$ phosphorylation under Lapatinib treatment also resulted in absence of transient memory in polarized morphology after stimulus removal (*Figure 2I*). This establishes a direct link between memory in polarized receptor activity and memory in polarized cell shape.

## Transient memory in cell polarization is translated to transient memory in directional migration

To test the phenotypic implications of the transient memory in cell polarization, we analyzed the motility features of the engineered $MCF7 - EGFR^{mCitrine}$, as well as of MCF10A cells at physiological EGF concentrations. Cells were subjected to a 5 hr dynamic EGF$^{647}$ gradient that was linearly distributed within the chamber, with EGF$^{647}$ ranging between 25-0 ng/ml, allowing for optimal cell migration (*Figure 3—figure supplement 1A, B*). The gradient steepness was progressively decreased in a controlled manner, rendering an evolution towards a ~50% shallower gradient over

time (*Figure 3—figure supplement 1B*). Automated tracking of single-cell's motility trajectories was performed for 14 hr in total. $MCF7 - EGFR^{mCitrine}$, as well as MCF10A cells migrated in a directional manner toward the EGF$^{647}$ source (*Figure 3A*- and *Figure 3—figure supplement 1C, D* - left, green trajectory parts). This directed migration persisted for transient period of time after the gradient wash-out (*Figure 3A*- and *Figure 3—figure supplement 1C, D* - left, red trajectory parts, *Figure 3—video 1*), indicating that cells maintain memory of the location of previously encountered source. After the memory phase, the cells transitioned to a migration pattern equivalent to that in the absence of a stimulus (*Figure 3A* right, *Figure 3—figure supplement 1C, D* middle). Uniform stimulation with 20 ng/ml EGF$^{647}$ did not induce directed migration in either of the cell lines, although the overall migration distance was increased in accordance with previous findings (*Brüggemann et al., 2021*; *Figure 3—figure supplement 1C, D*, right). Quantification of the directionality of single cells' motion, that is defined as the displacement over travelled distance, showed that for MCF10A cells, it was significantly higher during the gradient stimulation (5 hr) as compared to no- or uniform-stimulation case (*Figure 3B*). Moreover, the directionality estimated in the 9 hr time-frame after the gradient removal was greater than the one in continuous stimulus absence, corroborating that cells transiently maintain memory of the previous direction of migration.

This was also reflected in the projection of the cell's relative displacement angles ($\cos\theta$) estimated along the gradient direction ($\pi$) at each time point (*Figure 3—figure supplement 2A*), representing the angular alignment of the cells to the source direction. The cellular migration trajectories aligned with the source direction ($\cos\theta$ approached 1) during, and maintained this temporally after gradient removal, before returning to a migration pattern characteristic for stimulus absence or during uniform stimulation ($\cos\theta \sim 0$, *Figure 3C* top, *Figure 3—figure supplement 2B*). Calculating the similarity between the kernel density distribution estimate (KDE) of the angular alignment distributions at each point in the gradient series with that in continuous stimulus absence, showed that the distributions approach each other only $\sim$ 50 min after the gradient removal (*Figure 3C*, bottom; *Figure 3—figure supplement 2C*). Additionally, the calculated similarity between the KDE distributions during the gradient (5 hr) and the $50 min$ memory period further corroborated this finding (*Figure 3—figure supplement 2C*). The average memory phase in directional motility thus corresponds to the time-frame in which the memory in polarized $EGFR^{mCitrine}$ phosphorylation and cell shape is maintained (*Figures 2E and 3C*), indicating that the metastable signaling state is translated to a stable prolonged directed migration response after gradient removal.

To investigate whether the motility patterns during the gradient and the memory phase have equivalent characteristics, we fitted the motility data using a modified Ornstein-Uhlenbeck process (*Uhlenbeck and Ornstein, 1930*; *Svensson et al., 2017*) and used the extracted migration parameters to generate synthetic single-cell trajectories (Materials and methods). In absence of stimulus, the cellular motion resembled a random walk process (RW: *Figure 3D* right, *Figure 3—figure supplement 2D, E* middle), persistent random walk (PRW) was characteristic for the uniform stimulation case (*Figure 3—figure supplement 2D, E* right), whereas biased PRW described the migration in gradient presence (PBRW, *Figure 3D*- and *Figure 3—figure supplement 2D*, left, green trajectory part). Extending the bias duration during the interval of the experimentally observed memory phase (PB(t)RW) was necessary to reproduce the transient persistent motion after gradient removal (*Figure 3D*- and *Figure 3—figure supplement 2D*, left, blue trajectory part; *Figure 3E and F*; *Figure 3—figure supplement 2F*).

To corroborate the link between memory in polarized receptor activity, memory in polarized cell shape and memory in directional migration, we also quantified the directional migration of MCF10A cells when subjected to Lapatinib during gradient wash-out (*Figure 3G*). The directionality after gradient removal was significantly lower than in the case without Lapatinib (*Figure 3H*), suggesting that cells rapidly switch to a RW migration pattern upon gradient wash-out due to the absence of memory in polarized $EGFR^{mCitrine}$ phosphorylation (*Figure 2G,I*). Thus, single-cell motility trajectories that closely resembled the experimentally observed ones could be mimicked with the PB(t)RW simulation, where the bias duration corresponded to the duration of the gradient (*Figure 3—figure supplement 2E* left, **2G**). Quantification of the average cells' relative displacement angles showed as well that $\cos\theta$ approaches 0 exponentially after gradient removal (*Figure 3I*, *Figure 3—figure supplement 2G*), suggesting that majority of cells display absence of memory in directional migration under Lapatinib treatment.

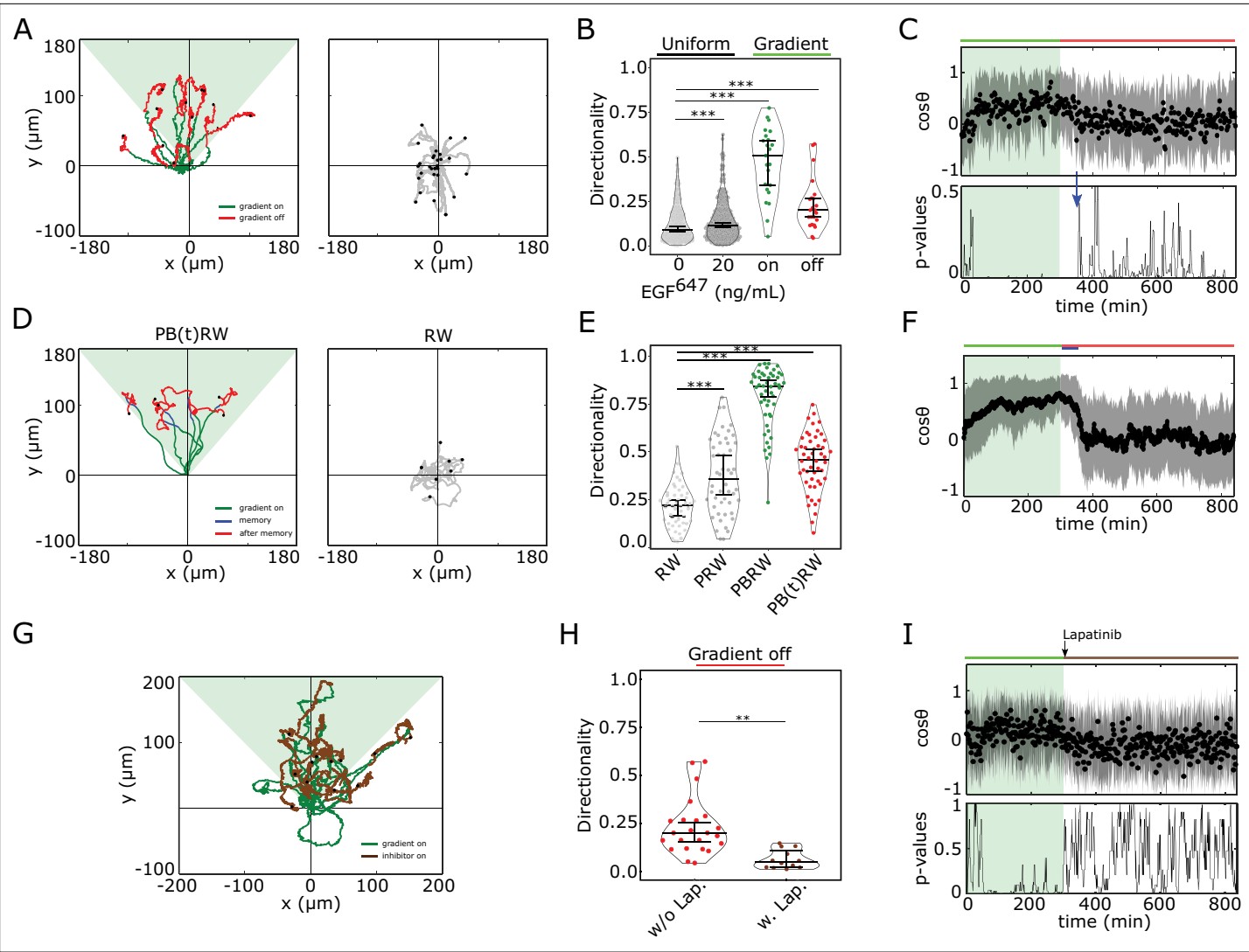

**Figure 3.** Cells display memory in directional migration toward recently encountered signals. (**A**) Left: representative MCF10A single-cell trajectories. Green - 5 hr during and red line - 9 hr after dynamic EGF$^{647}$ gradient (shaded). Exemplary cell in *Figure 3—video 1*. Right: Same as in (**A**), only 14 hr in continuous EGF$^{647}$ absence. Black dots: end of tracks. (**B**) Directionality (displacement/distance) in MCF10A single-cell migration during 14 hr absence (0 ng/ml; n=245, N=3) or uniform 20 ng/ml EGF$^{647}$ stimulation (n=297, N=3); 5 hr dynamic EGF$^{647}$ gradient (green) and 9 hr during wash-out (red; n=23, N=5). p-values: ***p≤ 0.001, two-sided Welch's t-test. Error bars: median±95%C.I. (**C**) Top: Projection of the cells' relative displacement angles (mean ± s.d.; n=23, N=5) during (green shaded) and after 5 hr dynamic EGF$^{647}$ gradient. Green/red lines: stimulus presence/absence. Bottom: Kolmogorov-Smirnov (KS) test p-values depicting end of memory in directional migration (blue arrow, $t = 350 min$). KS-test estimated using 5 time points window. For (**A-C**), data sets in *Figure 3—figure supplements 1D and 2A*. (**D**) Representative in silico single-cell trajectories. Left: PB(t)RW: Persistent biased random walk, bias is a function of time (green/blue trajectory part - bias on). Right: RW: random walk. (**E**) Corresponding directionality estimates from n=50 realizations, data in *Figure 3—figure supplement 2D*. PRW: persistent random walk. p-values: ***p≤ 0.001, two-sided Welch's t-test. Error bars: median±95%C.I. (**F**) Same as in (**C**), top, only from the synthetic PB(t)RW trajectories. (**G**) MCF10A single-cell trajectories quantified 5 hr during (green) and 9 hr after (brown) dynamic EGF$^{647}$ gradient (shading) wash-out with 3 µM Lapatinib. n=12, N=5. See also *Figure 3—video 2*. (**H**) Directionality in single-cell MCF10A migration after gradient wash-out with (brown, n=12, N=5) and without Lapatinib (red, n=23, N=5). p-values: **p≤ 0.01, KS-test. Error bars: median±95%C.I. (**I**) Same as in (**C**), only for the cells in (**G**). See also *Figure 3—figure supplement 2H*.

The online version of this article includes the following video, source data, and figure supplement(s) for figure 3:

**Source data 1.** Source data for *Figure 3*.

**Figure supplement 1.** Characterization of $MCF7 - EGFR^{mCitrine}$ and MCF10A single-cell migration.

**Figure supplement 1—source data 1.** Source data for *Figure 3—figure supplement 1*.

**Figure supplement 2.** Characterization of single cell migration patterns.

**Figure supplement 2—source data 1.** Source data for *Figure 3—figure supplement 2*.

*Figure 3 continued on next page*

*Figure 3 continued*

**Figure supplement 3.** Quantifying duration of memory in directional migration from single-cell $\cos\theta$ profiles.

**Figure supplement 3—source data 1.** Source data for *Figure 3—figure supplement 3*.

**Figure 3—video 1.** Corresponding to Figure 3A.

https://elifesciences.org/articles/76825/figures#fig3video1

**Figure 3—video 2.** Corresponding to Figure 3G.

https://elifesciences.org/articles/76825/figures#fig3video2

In order to dissect better the cell-to-cell variability in this case, we also calculated memory duration from single cell $\cos\theta$ profiles. For this, single-cell trajectories were first smoothed using Kalman filter (Materials and methods). The quantification showed that majority of the cells displayed absence of or shorter memory in directional migration, with a mean value of ~25 min (*Figure 3—figure supplement 3A, B, D*). Since under Lapatinib treatment, EGFR phosphorylation rapidly decays (*Figure 2G*), this residual memory in some cells likely results from memory in cytoskeletal asymmetries, as previously suggested (*Prentice-Mott et al., 2016*). Without Lapatinib treatment however, the duration of memory estimated from single-cell $\cos\theta$ profiles was of the order of 90 min (*Figure 3—figure supplement 3A, C, E*). If we therefore account in this case also the contribution of cytoskeletal memory, then the memory in directional migration which results from memory in polarized EGFR phosphorylation is on average ~50 min, similar to the deduced values from the single-cell kymograph quantification (*Figure 2E*).

## Molecular working memory enables cells to navigate in dynamic chemoattractant fields

To test whether the identified memory enables cellular navigation in environments where signals are disrupted but also change over time and space, we subjected cells in the simulations and experiments to a changing growth factor field. The field was generated by a sequence of signals, starting with a dynamic gradient whose steepness changed over time, and was temporary disrupted for a time interval shorter than the interval of memory in cell polarization. This was followed by a second static gradient in the same direction, that after an equivalent disruption period was followed by a third dynamic gradient in the opposite direction (*Figure 4A*). The in silico migration simulations showed that the cell can sense the initial dynamic gradient and polarizes in the direction of maximal attractant concentration, resulting in directed migration (*Figure 4B*, *Figure 4—figure supplement 1A*, *Figure 4—video 1*). The simulations also predicted that the memory of the previously encountered signal localization enables maintaining robust directional migration even when the signal was disrupted, while still remaining sensitive to the newly emerging signal from the opposite direction. The in silico cell rapidly adapted the orientation when encountering the third signal, demonstrating that the proposed mechanism can also account for prioritizing newly encountered signals. Such a dynamic memory which enables information of previous signals to be temporally maintained while retaining responsiveness to upcoming signals, and thereby manipulate the stored information, in neuronal networks is described as a working memory (*Atkinson and Shiffrin, 1968*).

If the signal disruption is however longer than the duration of the working memory, the simulations demonstrated that cells cannot integrate the signals. In turn, cells respond to each signal individually, as the directional migration after the memory is lost, resulting in a shorter range migration trajectory (*Figure 4—figure supplement 1B*, *Figure 4—video 2*). On the other hand, if the system has a long-term memory, as resulting from organization in the stable polarized regime, the simulations showed that cellular adaptation to a changing gradient field is hindered (*Figure 4—figure supplement 1C,D Figure 4—video 3*). The initial dynamic gradient shifted the system to the stable polarization steady state where it was maintained on a long-term, such that sensitivity to upcoming signals from the same direction was hindered. Even more, the cell could not resolve the conflicting information from a subsequent gradient from the opposite direction, as the signals induced high receptor activity on the opposed cell sides, resulting in halted migration. These results therefore highlight the importance of working memory for generating memory-guided migration over long trajectories.

We next tested these predictions experimentally by establishing an equivalent dynamic EGF[647] spatial-temporal field in a controlled manner in the microfluidic chamber, and quantified the migratory

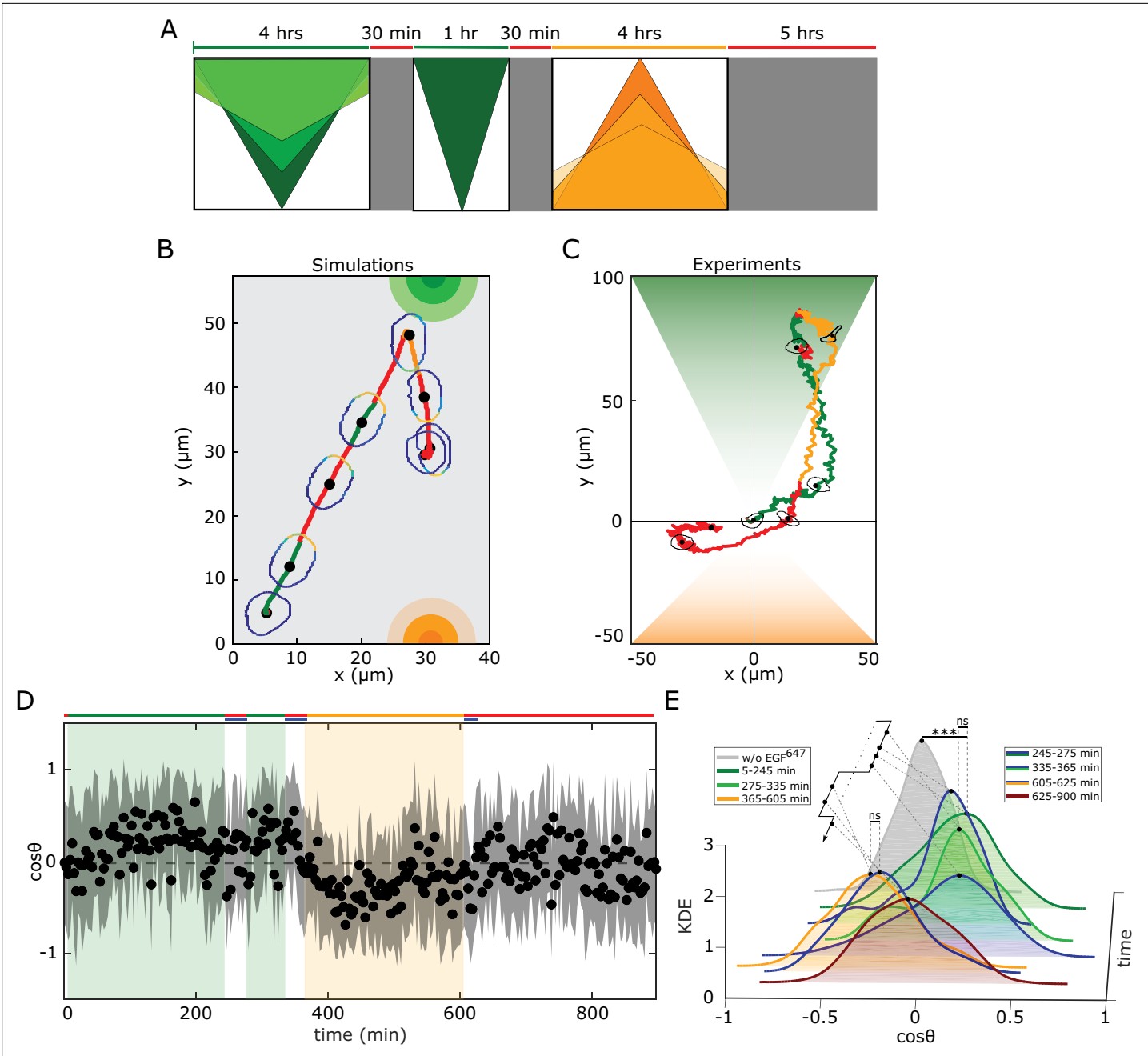

**Figure 4.** Working memory enables history-dependent single-cell migration in changing chemoattractant field. (**A**) Scheme of dynamic spatial-temporal growth factor field implemented in the simulations and experiments. Green(orange)/red: gradient presence/absence. (**B**) In silico cellular response to the sequence of gradients as depicted in (**A**), showing changes in EGFR activity, cellular morphology and respective motility trajectory over time. Trajectory color coding corresponding to that in (**A**), cell contour color coding with respective $E_p$ values as in *Figure 1E*. Cell size is magnified for better visibility. See also *Figure 4—figure supplement 1A*, *Figure 4—video 1*. (**C**) Representative MCF10A single-cell trajectory and cellular morphologies at distinct time-points, when subjected to dynamic EGF[647] gradient field as in (**A**) (gradient quantification in *Figure 4—figure supplement 1E*). Trajectory color coding corresponding to that in (**A**). See also *Figure 4—video 4*. Full data set in *Figure 4—figure supplement 1F*. (**D**) Projection of cells' relative displacement angles ($\cos\theta$) depicting their orientation towards the respective localized signals. Mean ± s.d. from n=12, N=5 is shown. (**E**) Corresponding kernel density estimates (intervals and color coding in legend). p-values: ***, p≤0.001, ns: not significant, KS-test.

The online version of this article includes the following video, source data, and figure supplement(s) for figure 4:

**Source data 1.** Source data for *Figure 4*.

**Figure supplement 1.** Single-cell navigation in changing growth factor fields.

*Figure 4 continued on next page*

*Figure 4 continued*

**Figure supplement 1—source data 1.** Source data for *Figure 4—figure supplement 1*.

**Figure supplement 2.** Resolving simultaneous signals with opposed localisation is optimal at criticality.

**Figure 4—video 1.** Corresponding to Figure 4B.
https://elifesciences.org/articles/76825/figures#fig4video1

**Figure 4—video 2.** Corresponding to Figure 4—figure supplement 1B.
https://elifesciences.org/articles/76825/figures#fig4video2

**Figure 4—video 3.** Corresponding to Figure 4—figure supplement 1C, D.
https://elifesciences.org/articles/76825/figures#fig4video3

**Figure 4—video 4.** Corresponding to Figure 4C.
https://elifesciences.org/articles/76825/figures#fig4video4

**Figure 4—video 5.** Corresponding to Figure 4—figure supplement 2A, B.
https://elifesciences.org/articles/76825/figures#fig4video5

**Figure 4—video 6.** Corresponding to Figure 4—figure supplement 2C, D.
https://elifesciences.org/articles/76825/figures#fig4video6

profile of MCF10A cells (*Figure 4—figure supplement 1E*). The MCF10A cells sensed the initial dynamic gradient field and migrated in the direction of increasing chemoattractant concentration, maintaining the directionality even when the signal was temporary disrupted. Despite the memory in cell polarization, cells remained responsive and adapted the duration of directional migration when presented with a second static gradient from the same direction, and subsequently prioritized the third, newly encountered signal with opposed orientation (exemplary trajectory in *Figure 4C*, *Figure 4—video 4*, *Figure 4—figure supplement 1F,G*). Thus, the predictions derived by the numerical simulations quantitatively captured that the proposed mechanism of navigation enables integration of, and adaptation to changes in signal localization. The distinction between the simulations and the experiments (*Figure 4B and C*) is only in the details of the migration pattern, since the PBRW migration mode was not included in the physical model of the cell for simplicity. The temporal memory in directional migration as well as the continuous adaptation of MCF10A cells to novel cues was also reflected in the projection of the cell's relative displacement angles (*Figure 4D*). The thereby derived KDE distributions during the first and second gradient (5–245 min; 275–335 min, respectively), as well as the corresponding intervals in which the gradient has been disrupted (245–275 min; 335–365 min, respectively) were statistically similar to each other, demonstrating that cells maintain the direction of migration in the intermittent intervals when the gradient was interrupted (*Figure 4E*). Moreover, these distributions statistically differed from the one characterizing cellular migration in continuous EGF$^{647}$ absence (w/o EGF$^{647}$, distribution symmetrically distributed around $\cos\theta = 0$). The presence of the third gradient from the opposite direction (365–605 min) on the other hand, induced a shift in the respective KDE distribution to negative $\cos\theta$ values, reflecting that cells revert the direction of migration (established in ~10 min). Furthermore, the reverse migration was maintained for approximately 20 min after wash-out of the third gradient (KDE 605–625 min). The statistical similarity between these two distributions demonstrates that cells also establish transient memory of the last detected signal, before reverting to a random walk migration mode (KDE 625–900 min similar to KDE w/o EGF$^{647}$). These results therefore demonstrate that cells utilize molecular working memory to navigate in changing gradient fields.

Navigation in non-stationary fields, however, also necessitates integration of information, requiring active comparison during migration task execution. We therefore tested next numerically whether the identified organization at criticality enables resolving simultaneous gradients with different amplitudes from opposite sides, that temporally vary in time. In the simulations, the cell sensed the presence of both signals, as reflected in the respective increase in EGFR phosphorylation. However, the net polarization towards the higher amplitude gradient was dominant, resulting in a clear directional migration toward this signal (*Figure 4—figure supplement 2A, B*). After the gradient removal, the EGFR phosphorylation and the cell shape remained transiently polarized, manifesting memory of the recently encountered stronger signal that was translated to memory in directional migration, before the cell reverted to a random walk migration (*Figure 4—video 5*). In contrast, if the system has a long-term memory as resulting from organization in the stable polarized state, the simulations showed that EGFR

phosphorylation increased almost equivalently with respect to both signals, despite the difference in signal amplitudes. This hindered the responsiveness of the cell such that migration could not be effectively exhibited (*Figure 4—figure supplement 1C, D*; *Figure 4—video 6*). These simulations therefore suggest that critical organization of receptor networks is in general crucial for performing complex cellular behavior that goes beyond simple stimulus-response associations.

## Discussion

Our data establishes that mammalian cells use a mechanism of working memory to navigate in complex environments where the chemical signals are disrupted or vary over time and space. Previous observations of memory in directed migration have been explained through the presence of bistable dynamics, where the transition from the basal to the polarized steady state and vice versa (after a memory phase) is regulated by two finely tuned thresholds. The authors however did not identify potential molecular elements that store this information, or regulate the thresholds (*Skoge et al., 2014*). Similarly, the remaining proposed models of polarization also rely on steady-state description of the basal and polarized states (*Levine et al., 2002*; *Mori et al., 2008*; *Goryachev and Pokhilko, 2008*; *Beta et al., 2008*; *Trong et al., 2014*), and thereby cannot account for the rapid adaptation to changes in signal localization.

The mechanism of transient memory we report here is realized on a molecular level by a prolonged polarized phosphorylation state of a receptor tyrosine kinase. Dynamically, this state emerges for organization at criticality, where a slow-escaping remnant from the polarized state or a dynamically metastable 'ghost' state is generated, and endows cells with robust transient maintenance of directional migration after signal removal. Although the observed memory in directional migration is in part supported by the memory in cytoskletal asymmetries as previously suggested (*Prentice-Mott et al., 2016*), the memory in receptor signaling we identify here provides a crucial bridge between the rapid receptor phosphorylation/dephosphorylation events and the long-range cellular migration. In particular, the organization at criticality endows the system with a slow time-scale through which the prolonged receptor phosphorylation state can be maintained on average for ~40–50 min after signal removal, which in turn maintains the polarized cell shape, and thereby directional migration in absence of a signal. Moreover, we have demonstrated that this memory arising from a metastable state uniquely ensures the ability of cells to quickly adapt to changes in the external environment.

Thus, our results suggest that in order to balance between a robust response and adaptation to novel signals, cell utilize an optimal receptor amount at the plasma membrane that corresponds to organization at criticality. The theoretical analysis suggest that the closeness of the receptor amount to the one corresponding to the critical transition is reflected in the memory duration. It can be therefore suggested that the observed variability in the experimentally identified memory length likely results from cell-to-cell variability in receptor concentration at the plasma membrane. Moreover, these results also suggest that a higher number of sensory units at the plasma membrane does not necessarily imply improved sensitivity of cells, but rather counterintuitively , leads to permanent memory of the initially encountered signal. This in turn will limit the cellular responsiveness to upcoming signal changes. It would be therefore of interest to study whether receptor networks are self-organized at criticality through an active sensing mechanism, or this feature has been fine-tuned through evolution, as a means for optimizing sensing and computational capabilities of cells.

Our work furthermore suggest that this general mechanism of a system poised at criticality can explain a wide range of biologically relevant scenarios, from the integration of temporally and spatially varying signals, to how extracellular information is transformed into guidance cues for memory-directed migration. Such memory-guided navigation is advantageous when migration must be realized over long and complex trajectories through dense tissues where the chemical cues are disrupted or only locally organized (*Lämmermann et al., 2013*). We have demonstrated here that the molecular working memory in cell polarization and therefore the capabilities of cells to navigate in a complex environment are an emergent feature of receptor networks.

## Materials and methods

**Key resources table**

| Reagent type (species) or resource | Designation | Source or reference | Identifiers | Additional information |
|---|---|---|---|---|
| Cell line (*Homo sapiens*) | MCF-7 | ECACC | Cat.No.86012803 | |
| Cell line (*Homo sapiens*) | MCF10A | ATCC | CRL-10317 | |
| Recombinant DNA reagent | EGFR-mCitrine | *Baumdick et al., 2015* | | |
| Recombinant DNA reagent | PTB-mCherry | *Fueller et al., 2015* | | |
| Recombinant DNA reagent | cCbl-BFP | *Fueller et al., 2015* | | |
| Peptide, recombinant protein | Fibronectin | Sigma-Aldrich | F0895-1MG | |
| Peptide, recombinant protein | Collagen | Sigma-Aldrich | C9791-50MG | |
| Chemical compound, drug | Lapatinib | Cayman chemicals | Cay11493-10 | |
| Chemical compound, drug | Hoechst 33,342 | Thermo Fisher Sc. | 62,249 | |
| Chemical compound, drug | Dulbecco's modified Eagle's medium (DMEM) | PAN Biotech | Cat. # P04-01500 | |
| Chemical compound, drug | MEM Amino Acids Solution (50 x) | PAN Biotech | Cat. # P08 32,100 | |
| Chemical compound, drug | Penicillin- Streptomycin | PAN Biotech | Cat. # P06 07100 | |
| Chemical compound, drug | Fetal Bovine Serum | Sigma-Aldrich | Cat. # F7524 | |
| Chemical compound, drug | EGF | Sigma-Aldrich | Cat. # E9644 | |
| Chemical compound, drug | Hydrocortisone | Sigma-Aldrich | Cat. #H-0888 | |
| Chemical compound, drug | Cholera toxin | Sigma-Aldrich | Cat. #C-8052 | |
| Chemical compound, drug | Insulin | Sigma-Aldrich | Cat. #I-1882 | |
| Chemical compound, drug | Horse Serum | Invitrogen | 26050088 | |
| Chemical compound, drug | FuGENE6 | Promega | E2691 | |
| Software, algorithm | Python | Python software foundation | RRID:SCR_008394 | |
| Software, algorithm | Matlab | MathWorks | RRID:SCR_001622 | |
| Software, algorithm | XPPAUT | http://www.math.pitt.edu/~bard/xpp/xpp.html | | |
| Software, algorithm | Trackmate | https://doi.org/10.1016/j.ymeth.2016.09.016 | | |
| Software, algorithm | Fiji, ImageJ | https://doi.org/10.1038/nmeth.2019 | | |
| Other | EGF-Alexa647 | *Sonntag et al., 2014* | Prof. Luc Brunsveld, University of Technology, Eindhoven | Methods |
| Other | Cellasic ONIX plates | Merck Chemicals | M04G-02-5PK | Methods |

## Cell culture

MCF7 cells (sex: female, ECACC, Cat. No. 86012803) were grown at 37 °C and 5% $CO_2$ in Dulbecco's Eagle's medium (DMEM) (PAN-Biotech, Germany), supplemented with 10% inactivated Fetal Calf Serum (FCS) (Sigma-Aldrich), 100 ng ml⁻¹ L-Glutamine, 0.5 mg ml⁻¹ non-essential amino acids, 100 µg ml⁻¹ penicillin and 100 µg ml⁻¹ streptomycin (PAN-Biotech, Germany). Serum starvation was performed by culturing the cells in DMEM supplemented with 0.5% FCS, 100 µg ml⁻¹ penicillin and 100 µg ml⁻¹ streptomycin (PAN-Biotech, Germany). MCF10A cells (sex: female, ATCC-CRL 10317) were grown at 37 °C and 5% $CO_2$ in Mammary Epithelial Cell Growth Basal medium (MEBM from Lonza Pharma & Biotech), supplemented with 5% Horse Serum (HS) (Invitrogen), 20 ng ml⁻¹ EGF (Sigma-Aldrich), 0.5 mg ml⁻¹ hydrocortisone (Sigma-Aldrich), 100 ng ml⁻¹ cholera toxin (Sigma-Aldrich), 10 µg ml⁻¹insulin (Sigma-Aldrich), 100 µg ml⁻¹ penicillin and 100 µg ml⁻¹ streptomycin. Serum starvation was performed by culturing the cells in the DMEM supplemented with 0.5% HS, 0.5 mg ml⁻¹ hydrocortisone (Sigma-Aldrich), 100 ng ml⁻¹ cholera toxin (Sigma-Aldrich), 100 µg ml⁻¹ penicillin and 100 µg ml⁻¹ streptomycin. MCF7 and MCF10A cells were authenticated by Short Tandem Repeat (STR) analysis and did not contain DNA sequences from mouse, rat and hamster (Leibniz-Institut DSMZ). Cells were regularly tested for mycoplasma contamination using MycoAlert Mycoplasma detection kit (Lonza).

## Transfection and cell seeding

For $EGFR^{mCitrine}$ polarization experiments, $2.5 \times 10^5$ MCF7 cells were seeded per well in a six-well Lab-Tek chamber (Nunc) until 80% confluence was reached. After 9–10 hr of seeding, transient transfection was performed with a total of 1 µg of plasmids ($EGFR^{mCitrine}$, $PTB^{mCherry}$ and $cCbl^{BFP}$ at ratio 4:3:4 by mass) using FUGENE6 (Roche Diagnostics) transfection reagent and Opti-MEM (Gibco - Thermo Fisher Scientific) according to manufacturer's procedure. All plasmids were generously provided by Prof. P. Bastiaens, MPI of Molecular Physiology, Dortmund. Cells were incubated for 7–8 hr to allow the expression of the transfected proteins prior to experiments. To detach the cells, the growth media was discarded and cells were washed once with DPBS (PAN Biotech) before adding 100 µl Accutase (Sigma-Aldrich). After 10 min incubation period at 37 °C and 5% $CO_2$, fresh growth media was added, and the cell density and viability was measured using cell counter (Vi-CELL XR Cell Viability Analyzer System). After spinning down, the cells were diluted to $10 \times 10^6$ cells/ml. The M04-G02 microfluidic gradient plates (Merck Chemicals) were primed for usage by flowing cell culture growth media through the cell chamber for 5 min and cells were subsequently seeded according to manufacturer's instructions.

For migration experiments with uniform $EGF^{647}$ stimulation, 6-well Lab-Tek plates were coated with Collagen (Sigma-Aldrich) in 0.1 M Acetic acid (Sigma-Aldrich) for MCF7 (100 µg cm⁻²), and Fibronectin (Sigma-Aldrich) in Phosphate-Buffered Saline (DPBS) (PAN-Biotech) for MCF10A cells (2 µg ml⁻¹), and stored in incubator at 37 °C overnight for evaporation. Excessive media was removed and the wells were washed with DPBS before seeding cells. MCF7 cells were seeded and transfected as described above. In the case of MCF10A cells, $1 \times 10^5$ cells per well were used for seeding. For migration experiments with gradient $EGF^{647}$ stimulation, MCF7 cells were transferred to the coated M04-G02 microfluidic gradient plates as described above. Before seeding, MCF10A cells were detached from 6 well Lab-Teks by discarding the growth media and washing once with DPBS (PAN Biotech) before adding 100 µl Accutase (Sigma-Aldrich). After 20–30 min incubation period at 37 °C and 5% $CO_2$, fresh cell growth media was added, and the cell density and viability were measured using a cell counter (Vi-CELL XR Cell Viability Analyzer System). After spinning down, the cells were diluted to $2 \times 10^6$ cells/ml, and subsequently seeded in the microfluidic plates according to manufacturer's instructions.

## Reagents

For gradient quantification, Fluorescein (Sigma Aldrich) was dissolved in Dulbecco's modified Eagle's medium (with 25 mM HEPES, without Phenol Red) (PAN Biotech). Imaging media: DMEM without Phenol Red was mixed with 25 mM HEPES. For nuclear staining, 20 mM Hoechst 33342 (Thermo Fisher Scientific) was mixed with DPBS and diluted to 2 µM working concentration. EGFR inhibitor Lapatinib (Cayman Chemical, Ann Arbor, MI) was solubilized in DMSO (Thermo Fisher Scientific) to a stock concentration of 5 mM and stored at –20 °C.

## Confocal and wide-field microscopy

Confocal images were recorded using a Leica TCS SP8i confocal microscope (Leica Microsystems) with an environment-controlled chamber (Life Imaging Services) maintained at 37 °C and HC PL APO 63 x/1.2 N.A / motCORR CS2 water objective (Leica Microsystems) or a HC PL FLUOTAR 10 x/0.3 N.A. dry objective (Leica Microsystems). mCitrine, mCherry and Alexa647 were excited with a 470nm–670nm pulsed white light laser (Kit WLL2, NKT Photonics) at 514 min, 561 nm, and 633 nm, respectively. BFP and Hoechst 33342 (Thermo Fisher Scientific) were excited with a 405 nm diode laser. The detection of fluorescence emission was restricted with an Acousto-Optical Beam Splitter (AOBS): BFP (425nm–448nm), Hoechst 33342 (425nm–500nm), mCitrine (525nm–551nm), mCherry (580nm–620nm), and Alexa647 (655nm–720nm). Transmission images were recorded at a 150–200% gain. To suppress laser reflection, Notch filter 488/561/633 was used whenever applicable. When using the dry objective for migration experiments, the pinhole was set to 3.14 airy units and 12-bit images of 512 × 512 pixels were acquired in frame sequential mode with 1 x frame averaging. When using the water objective for polarization experiments, the pinhole was fixed (1.7 airy units) for all channels. The Leica Application Suite X (LAS X) software was used.

Wide field images were acquired using an Olympus IX81 inverted microscope (Olympus Life Science) equipped with a MT20 illumination system and a temperature controlled $CO_2$ incubation chamber at 37 °C and 5% $CO_2$. Fluorescence and transmission images were collected via a 10 x/0.16 NA air objective and an Orca CCD camera (Hamamatsu Photonics). Hoechst 33342 fluorescence emission was detected between 420nm–460nm via DAPI filter, mCitrine fluorescence emission between 495nm–540nm via YFP filter and Alexa647 fluorescence emission between 705nm–745nm via Cy5 filter. The xCellence (Olympus) software was used.

## Gradient establishment for polarization and migration experiments

The CellAsic Onix Microfluidic Platform (EMD Millipore) was used for gradient cell migration and $EGFR^{mCitrine}$ phosphorylation polarization experiments. For $EGFR^{mCitrine}$ phosphorylation polarization experiments, 1 hr gradient stimulation was established using CellASIC ONIX2 software as follows. (i) Pre-stimulus: Imaging media was flowed from well groups 3 and 4 (CellAsic Onix Manual - https://www.merckmillipore.com/) at low pressure (2.5 kPa) for 5 min. (ii) Gradient establishment: After closing well group 3, pre-loaded EGF$^{647}$ (10 ng mL$^{-1}$) was flowed through well group 2 and imaging media from well group 4 at high pressure (15 kPa) for 15 min (iii) Gradient maintenance: The pressure was reduced to 10 kPa for 45 min. (iv) Washout: After closing well groups 2 and 4, imaging media was flowed from well groups 3 and 5 at high pressure (15 kPa) for 15 min and maintained at low pressure (7 kPa) for 165 min. For single gradient migration experiments, this protocol was modified as follows: in step (iii), gradient maintenance was done for 285 min. In step (iv), maintenance was at low pressure for 585 min. 30 ng mL$^{-1}$ EGF$^{647}$ was used. For polarization experiments with inhibitor, the same protocol as for polarization experiments was used, except well group 3 and 5 were filled with 1 µM Lapatinib solution and in step (i) well group 3 was kept closed. For single cell gradient migration experiment with inhibitor, 3 µM Lapatinib was used.

For migration experiments under subsequent gradient stimuli / gradient quantification, the following changes in the steps were used: (ii) well group 2 with 30 ng mL$^{-1}$ EGF$^{647}$/ 2.5 µM Fluorescein was used. (iii) The gradient maintenance was done for 225 min. (iv) Washout: imaging media was flowed from well groups 3 and 4 at high pressure (15 kPa) for 15 min and maintained at low pressure (7 kPa) for 15 min. (v) Second gradient establishment: After closing well group 3, EGF$^{647}$ (30 ng ml$^{-1}$)/ 2.5 µM Fluorescein was flowed from well group 2 and imaging media from well group 4 at high pressure (15 kPa) for 15 min. (vi) The second gradient formed was maintained by reducing the pressure to 10 kPa for 45 min. (vii) Washout: imaging media was flowed from well groups 3 and 4 at high pressure (15 kPa) for 15 min and maintained at low pressure (7 kPa) for 15 min. (viii) Third gradient establishment: After closing well group 4, EGF$^{647}$ (30 ng mL$^{-1}$) / 2.5 µM Fluorescein was flowed from well group 5 and imaging media from well group 3 at high pressure (15 kPa) for 15 min. (ix) The third reversed gradient was maintained by reducing the pressure to 10 kPa for 225 min. (x) Washout: imaging media was flowed from well groups 3 and 4 at high pressure (15 kPa) for 15 min and maintained at low pressure (7 kPa) for 285 min.

## Imaging $EGFR^{mCitrine}$ phosphorylation polarization and single cell migration

Transfected $MCF7 - EGFR^{mCitrine}$ cells transferred to M04G-02 gradient plates as described above were incubated for at least 3 hr, followed by serum starvation for at least 6 hr before imaging.

Existing cell media was substituted right before imaging with imaging media. Confocal imaging for multiple positions at 1 min time interval using adaptive auto-focus system and the water objective was performed concurrently during the duration of the experiment using the Leica TCS SP8i.

For migration experiments under uniform EGF$^{647}$ stimulation, confocal laser scanning microscopy / transmission imaging of live $MCF7 - EGFR^{mCitrine}$ / MCF10A cells was done on a Leica TCS SP8i or Olympus IX81 for multiple positions at 3 min and 2 min time interval respectively, using the 10 × dry objective for 14 hr.

## EGF$^{647}$ / Fluorescein gradient quantification

hEGF$^{647}$ was generated in the lab of Prof. P. Bastiaens, MPI of molecular Physiology, Dortmund, using the His-CBD-Intein-(Cys)-hEGF-(Cys) plasmid (*Sonntag et al., 2014*), kindly provided by Prof. Luc Brunsveld, University of Technology, Eindhoven. Human EGF was purified from *E. coli* BL21 (DE3), N-terminally labeled with Alexa647-maleimide as described previously (*Sonntag et al., 2014*) and stored in PBS at –20 °C. To quantify the spatial extent of the EGF$^{647}$ /Fluorescein gradient, gradients were generated following the protocol described in sub-section 5.6 in plates without cells or matrix coating. Confocal images of Alexa647 /GFP channel were acquired at 1 min interval. A rectangular region of interest (including the perfusion channels and the culture chamber) was used to obtain an averaged pixel intensity profile using FIJI at each time point. This spatial profile was averaged across multiple experiments and then scaled with the mean intensity value in the perfusion channel, which corresponds to the applied EGF$^{647}$ /Fluorescein concentration.

## Quantifying $EGFR^{mCitrine}$ phosphorylation in single cells

To quantify plasma membrane $EGFR^{mCitrine}$ phosphorylation in live $MCF7 - EGFR^{mCitrine}$ cells, single-cell masks were obtained from the $EGFR^{mCitrine}$ channel at each time-point using FIJI (https://imagej.net/Fiji). All pixels within the obtained boundary were radially divided into two segments of equal areas (*Stanoev et al., 2018*), and the outer segment was taken to represent the plasma membrane. For the kymograph analysis, at each time point, the plasma membrane segment was divided into 4 quadrants in anti-clockwise direction, and each was divided into 5 spatial bins (*Figure 2A*). The fraction of phosphorylated $EGFR^{mCitrine}$ in each bin,   was estimated as:

$$EGFR_p^i(t) = \frac{PTB_{PM}^i(t)/(PTB_T(t) - PTB_{endo}(t))}{EGFR_{PM}^i(t)/EGFR_T(t)} \tag{1}$$

where $PTB_{PM}^i(t)$ and $EGFR_{PM}^i(t)$ are respectively the $PTB^{mCherry}$ and $EGFR^{mCitrine}$ fluorescence at $i^{th}$ plasma membrane bin, $PTB_T(t)$ and $EGFR_T(t)$ - respective total fluorescence in the whole cell, $PTB_{endo}(t)$ – the $PTB^{mCherry}$ fluorescence on vesicular structures in the cytoplasm. Endosomal structures were identified from the cytosol by intensity thresholding (1.5 s.d. percentile) and $PTB^{mCherry}$ fluorescence from these structures was subtracted from the $PTB_T(t)$, to correct for the $PTB^{mCherry}$ fraction bound to the phosphorylated $EGFR^{mCitrine}$ on endosomes.

Temporal profile of the fraction of phosphorylated $EGFR^{mCitrine}$ on the plasma membrane was obtained using:

$$EGFR_p(t) = \frac{\frac{\sum_{i=1}^{20} PTB_{PM}^i(t)}{(PTB_T(t) - PTB_{endo}(t))}}{\frac{\sum_{i=1}^{20} EGFR_{PM}^i(t)}{(EGFR_T(t))}} \tag{2}$$

and then normalized as:

$$EGFR_p(t) = \frac{EGFR_p(t) - <EGFR_p>_{t \in [0,5min]}}{max_t(EGFR_p(t)) - <EGFR_p>_{t \in [0,5min]}} \tag{3}$$

with <> being the temporal average in the pre-stimulation interval $t \in [0, 5min]$. The fraction of liganded receptor was calculated using:

$$EGF - EGFR(t) = \frac{EGF_{PM}}{EGFR_{PM}}(t) \tag{4}$$

To classify single cells into non-activated, activated (polarized $EGFR^{mCitrine}$ phosphorylation) and pre-activated (uniformly distributed $EGFR^{mCitrine}$ phosphorylation) upon gradient $EGF^{647}$ stimulation (*Figure 2—figure supplement 2A, B*), the following method was applied. To identify pre-activated cells, a Gaussian Mixture Model (GMM) was fitted to the histogram of $(EGFR_p^i)_{t\in[0,5min]}$ values from all the analysed cells, and the intersection point between the two normal distributions was identified. If more than 30% of the $(EGFR_p^i)_{t\in[0,5min]}$ pixel intensity values for any cell lie above the intersection point, the cell is classified as pre-activated. To distinguish between the non-activated and activated cells in the remaining population, average $EGFR^{mCitrine}$ phosphorylation value ($EGFR_p$) per cell was estimated during the pre-stimulation ($t\in[0,5min]$) and the stimulation period ($t\in[5min,65min]$) ($<EGFR_p>_{t\in[0,65]}$) from the temporal $EGFR^{mCitrine}$ phosphorylation profiles. Histogram of the respective $EGFR_p$ values was again fitted with a GMM model. All cells with an average $<EGFR_p>_{t\in[0,65]}$ value lying below the intersection point were considered to be non-activated, whereas those above - activated.

The average of the spatial projection of the fraction of phosphorylated $EGFR^{mCitrine}$ from single-cell kymographs (*Figure 2—figure supplement 1C*) was generated from the 20 cells that were polarized in the direction of the $EGF^{647}$ gradient. For each cell, a temporal average of $EGFR_p$ per bin was calculated for the duration of the gradient ($t\in[5min,65min]$) and the bin with the maximal $EGFR_p$ value was translated to $\pi$. The profiles were then smoothened using a rolling average with a window of 7 bins. The resulting profiles were then averaged over all cells and mean ± s.d. is shown.

The local spatial $EGF^{647}$ distribution around single cells (*Figure 2—figure supplement 1F*) was estimated as follows: the cell mask obtained using the $EGFR^{mCitrine}$ images were dilated outwards by 8 pixels to account for possible ruffles, and then by additional 15 pixels. The secondary rim of 15 pixels around the cell mask was used to calculate the spatial distribution of $EGF^{647}$ outside single cells. This outer contour was divided in 20 bins as for the kymographs, and $EGF^{647}$ intensity was quantified in each bin. The angle between the direction of $EGF^{647}$ and the direction of EGFR phosphorylation was calculated as the amount of radial bins between the maxima in the spatial projections. This bin-distance was then translated into an angle under the assumption of a circular perimeter.

In order to identify the characteristic features of the $EGFR^{mCitrine}$ phosphorylation profile during the transition from polarized to unpolarized state, the single-cell $EGFR_p(t)$ profiles with and without Lapatinib treatment after gradient wash-out were fitted to an inverse sigmoid function given by,

$$f(t) = \frac{a_0}{a^n + t^n} \qquad (5)$$

were $a_0$, $a$ are constants and $n$ is the Hill-coefficient (examples in *Figure 2—figure supplement 2E, F*). Non-linear least square method (python package *curve fit*) was used to perform the fitting. Under normal conditions (w/o Lapatinib), $a \sim 10$, $a_0 \sim 10^3$ and $n \sim 2.88$ fitted well the data ($R^2 \sim 0.79$). The same function however could not describe the EGFRp profiles in the Lapatinib treatment experiment (median $R^2 \sim 0.33$). The Lapatinib treatment profiles were therefore fitted by fixing $a = 10$, and leaving $a_0$ and $n$ as free parameters, as they determine the upper plateau and the steepness of the drop to the basal level. In this case, $a_0 \sim 19$ and $n \sim 1.28$ were identified from the fitting (median $R^2 \sim 0.84$, *Figure 2—figure supplement 2E, G*). From the fitted profiles in both cases, half-life was estimated to be the time frame in which 50% of $EGFR^{mCitrine}$ phosphorylation is lost after $EGF^{647}$ removal.

## Estimating memory duration in $\mathrm{EGFR}^{mCitrine}$ phosphorylation polarization

The duration of memory in $EGFR^{mCitrine}$ phosphorylation polarization in single cells was estimated from the temporal profile of the fraction of plasma membrane area with high $EGFR^{mCitrine}$ phosphorylation during and after gradient removal (*Figure 2D and E*). For this, the single-cell kymographs were normalized to a maximal value of 1 using

$$EGFR_p^i(t) = \frac{EGFR_p^i(t) - <EGFR_p>_{t\in[0,5min]}}{max_t(EGFR_p(t)) - <EGFR_p>_{t\in[0,5min]}} \qquad (6)$$

yielding the value of phosphorylated $EGFR^{mCitrine}$ per bin per time point $t$. Using the mean of $EGFR_p$ + s.d. over the whole experiment duration as a threshold, all $EGFR_p^i(t)$ lying above the threshold were taken to constitute the area of polarized $EGFR^{mCitrine}$ phosphorylation. To account for different

bin sizes, at each timepoint, the area of all bins with $EGFR_p$ above the threshold was summed and divided by the respective total cell area, yielding the temporal evolution of the fraction of polarized cell area (FPA) (*Figure 2D*). The end of the memory duration per cell was identified as the time point at which $FPA_{per-cell} < (FPA_{average} - s.d.)$ in 3 consecutive time points (*Figure 2E*).

## Quantifying morphological changes in response to EGF[647] in experiments and simulations

Morphological changes of polarized cells were quantified using the solidity (*Figure 2H1*) of each cell at each time point and the directed protrusive area towards and away from the gradient (*Figure 1G and H*; *Figure 2—figure supplement 2H*). The solidity $\sigma$ is the ratio between the cell's area $A_{cell}$ and the area of the convex hull $A_{convex}$ ($\sigma = \frac{A_{cell}}{A_{convex}}$). The memory duration in cell morphology was calculated from the single-cell solidity profiles, and corresponds to the time-point at which the solidity is below mean-s.d. estimated during gradient presence. The directed cell protrusion area was estimated by comparing single cell masks at two consecutive time points. To reduce noise effects, the masks were first subjected to a 2D Gaussian filtering using the *filters.gaussian* function from the *scipy* python package. Protrusions were considered if the area change was greater than 10 pixels or 1.2 µm$^2$ per time point. The front and the back of the cells were determined by identifying an axis that runs perpendicular to the gradient and through the cell nucleus of the initial time point. The directed cell protrusion area was then obtained using $\frac{A_{prot,front}}{A_{front}} - \frac{A_{prot,back}}{A_{back}}$. The final profiles of directed protrusive area were smoothed using 1D Gaussian filtering with the *filters.gaussian_filter1d* function from the *scipy* python package. For the equivalent quantification from the simulations, the same procedures were applied without an area threshold. The memory duration was estimated as the time point at which the directed protrusive area crosses zero after the gradient removal.

## Quantification of single-cell migration and duration of memory in directed cell migration

Single cell migration trajectories were extracted using Trackmate (*Tinevez et al., 2017*) in Fiji (*Schindelin et al., 2012*) using Hoechst 33342/transmission channel. From the positional information (x and y coordinates) of individual cell tracks, quantities such as Motility, Directionality and $\cos\theta$ were extracted using custom made Python code (Python Software Foundation, versions 3.7.3, https://www.python.org/). Directionality was calculated as displacement over total distance and statistical significance was tested using two-sided Welch's t-test. To quantify the memory duration in directed single-cell migration, the Kernel Density Estimate (KDE) from $\cos\theta$ quantification in the continuous absence of EGF[647] (uniform case, between 250–300 min) was compared with a moving window KDE (size of 5 time points) from the gradient migration profile, using two sided Kolmogorov-Smirnov test. To verify the absence of memory when cells were treated with Lapatinib during gradient wash-out, a moving window KDE (5 time points) from $\cos\theta$ obtained in this case was compared to the KDE in continuous absence of EGF[647] (uniform case *Figure 3—figure supplement 2B*, between 250–300 min) using two sided Kolmogorov-Smirnov test (*Figure 3I*). Furthermore, the KDE between 300–350 min and 350–840 min (after gradient removal) was statistically equivalent to the KDE in continuous absence of EGF[647], confirming the rapid switch from directed to random-walk migration in the Lapatinib case (*Figure 3—figure supplement 2H*). To estimate the time required for complete reversal of cell migration direction when the cells were subjected to a gradient from opposite direction, KDE distributions were compared between the following time windows: 275–335 min (second gradient), 335–365 min, 365–385 min, 375–385 min, and 365–605 min (third gradient).

To quantify the motility patterns of MCF10A cells in absence, uniform or gradient $EGF^{647}$ stimulation, we fitted the experimentally obtained single cell migration trajectories using modified Ornstein-Uhlenbeck process (mOU) (*Uhlenbeck and Ornstein, 1930*) that is defined by the Langevin equation for the velocity vector $\nu$:

$$\frac{d\nu(\mathbf{t})}{dt} = -\frac{1}{\tau} \cdot \nu(\mathbf{t}) + \frac{\sqrt{2D}}{\tau} \cdot (\xi(t) + b(t)) \tag{7}$$

where $\xi(t)$ represents a white noise component, D is a diffusion coefficient characteristic of a Brownian motion, $\tau$ is the persistence time and $b(t)$ models the contribution of the time-dependent bias. The experimental data was fitted to obtain values of D and $\tau$. In order to estimate D, Mean

Square Displacement (MSD) was calculated from the single cell tracks using $MSD(t) = <|\mathbf{x}_i(t) - \mathbf{x}_i(0)|^2>$, where $\mathbf{x}_i(t)$ is the tracked position of i-th cell in the 2D plane, $<>$ is the average across all single cell tracks, and |.| is the Euclidean distance (*Selmeczi et al., 2005*). To estimate D, the obtained MSD profile was fitted with a linear function (= 4Dt). Goodness of Fit for the different experimental conditions: 0 ng/ml EGF647, $R^2 = 0.975$; for uniform 20 ng/ml EGF647 stimulation, $R^2 = 0.995$. In order to estimate $\tau$, Velocity Auto-Correlation Function $VACF(t) = <\nu_i(t) \cdot \nu_i(0)>$, where $\nu_i(t)$ is the measured velocity of -th cell at time t, was fitted with a mono exponential function (= $\phi_0 \cdot e^{\frac{-t}{\tau}}$). Goodness of Fit: for 0 ng/ml EGF647 case - Standard Error of Estimate $SEOE = 0.0261$; for uniform 20 ng/ml EGF647 stimulation case, $SEOE = 0.0570$. Fitted values: for 0 ng/ml EGF647 case, $\tau = 11.105$, $D = 0.425$; for uniform 20 ng/ml EGF647 stimulation case, $\tau = 38.143$, $D = 2.207$; bias $b(t) = 0.134$.

To compute the duration of memory in directional migration after gradient removal for individual cells (*Figure 3—figure supplement 3*), single cell migration tracks were first smoothened using a Kalman-filter (python package *filterpy.kalman*) by predicting the cell position and velocity. The cell's displacement angles relative to the gradient direction ($\cos\theta$) were calculated for each cell at each timepoint, rendering single-cell $\cos\theta$ plots (*Figure 3—figure supplement 3B,C*). The memory duration was then calculated as the point where three consecutive timepoints in the $\cos\theta$ profiles fall below a threshold $\cos\theta$ value of 0.75.

## Reconstructing state-space trajectories from temporal $EGFR^{mCitrine}$ phosphorylation profiles

The state-space reconstruction in *Figure 2F and G* was performed using the method of time-delay. For a time series of a scalar variable, a vector $x(t_i)$, $i = 1, ...N$ in state-space in time $t_i$ can be constructed as following

$$\mathbf{X}(t_i) = [x(t_i), x(t_i + d), .., x(t_i + (m-1)d)] \qquad (8)$$

where $i = 1$ to $N - (m-1)d$, d is the embedding delay, $m$ - is a dimension of reconstructed space (embedding dimension). Following the embedding theorems by Takens (*Takens, 1980*; *Sauer et al., 1991*), if the sequence $X(t_i)$ consists of scalar measurements of the state of a dynamical system, then under certain genericity assumptions, the time delay embedding provides a one-to-one image of the original set, provided $m$ is large enough. The embedding delay was identified using the *timeLag* function (based on autocorrelation), the embedding dimension using the *estimateEmbeddingDims* function (based on the nearest-neighbours method), and the state-space reconstruction using the *buildTakens* function, all from the *nonlinearTseries* package in R (https://cran.r-project.org/web/packages/nonlinearTseries/index.html). Before state-space reconstructions, time series were smoothened using the *Savitzky-Golay* filter function in Python. For *Figure 2F*, $d = 26$, $d_e = 3$; for *Figure 2G*, $d = 50$, $d_e = 3$.

## Theoretical consideration of the navigation mechanism in a generalized reaction-diffusion signaling model

We consider a generalized form of a (mass-conserved) reaction-diffusion (RD) model of an $M$ ($\mathbf{U} \in \mathbf{R}^M$) component system in $N$ ($\mathbf{x} \in \mathbf{R}^N$) dimensional space

$$\frac{\partial \mathbf{U}(\mathbf{x}, t)}{\partial t} = \mathbf{F}(\mathbf{U}(\mathbf{x}, t)) + \mathbf{D} \cdot \nabla^2 \mathbf{U}(\mathbf{x}, t) \qquad (9)$$

where $\mathbf{F} \in \mathbf{R}^M$ is the reaction term, $\mathbf{D}$ is a $M \times M$ diagonal matrix of diffusion constants $D_j, j = 1, ..., M$, and $\nabla^2$ is the Laplacian operator. Standard analysis of such models relies on linear stability analysis to find the conditions for a Turing-type instability (*Turing, 1952*), such that the symmetric steady state becomes unstable and an asymmetric polarized state is stabilized. By its nature, the linear stability analysis makes no prediction about the transition process itself, and thereby the type of bifurcation that underlies it. To provide quantitative description of the symmetry breaking transition in reaction-diffusion models, local perturbation analysis can be applied (*Holmes et al., 2015*). However, this analysis is mainly restricted to models characterized with large diffusion discrepancy between the signaling components. The conditions for a pitchfork bifurcation (*PB*)-induced transition in a generic

RD model therefore have to be formally defined. Let $\mathbf{U_s} = (u_{is})$ for $i = 1, ..., M$, be the stable homogeneous symmetric steady state of the RD system. Consider a linear perturbation of the form

$$\mathbf{U}(\mathbf{x}, t) = \mathbf{U_s} + \delta\mathbf{U}(\mathbf{x})e^{(\lambda t)}, \quad \delta\mathbf{U}(\mathbf{x}) \in \mathbf{R}^M \tag{10}$$

where $\delta\mathbf{U}(\mathbf{x})$ is the spatial and $e^{(\lambda t)}$ is the temporal part of the perturbation. Substituting *Equation 10* in *Equation 9* yields a linearized eigenvalue equation whose solution can be determined by solving the characteristic equation, $F_\lambda = det(\lambda I_{M \times M} - J_{M \times M}) = 0$. J is the Jacobian matrix of the system defined by $J_{ij} = \frac{\partial F_i(\mathbf{U}(x,t))}{\partial U_j}, i = 1, ...., M, j = 1, ....., M$.

The system exhibits a *PB* if, an odd eigenfunction $\delta\mathbf{U}(\mathbf{x})$ such that $\delta\mathbf{U}(-\mathbf{x}) = -\delta\mathbf{U}(\mathbf{x})$, taken in the limit $\lambda \to 0$, fulfills the following condition (*Paquin-Lefebvre et al., 2020*):

$$\lim_{\lambda \to 0} F_\lambda = det(J) = 0. \tag{11}$$

When this conditions is satisfied, the symmetric, homogeneous steady state of the system undergoes a pitchfork bifurcation and an inhomogeneous steady state (IHSS) with two branches of asymmetric steady states emerges. In terms of polarization, these branches correspond to front-back-polarized states, where the orientation depends on the direction of the external signal (*Figure 1A*, *Figure 1—figure supplement 1A*).

To identify whether the PB is of sub-critical type, and thereby identify the presence of an $SN_{PB}$, a weakly nonlinear analysis of *Equation 9* must be performed to obtain description of the amplitude dynamics of the inhomogeneous state. This can be achieved using an approximate analytical description of the perturbation dynamics based on the Galerkin method (*Becherer et al., 2009*; *Rubinstein et al., 2012*; *Bozzini et al., 2015*). For simplicity, we outline the steps for a one-dimensional system ($N = 1$). As we are interested in the description of a structure of finite spatial size (i.e. finite wavelength $k$), the final solution of the PDE is expanded around the fastest growing mode, $k_m$ into a superposition of spatially periodic waves. That means that $u(x, t) \in \mathbf{U}$ can be written as:

$$u(x, t) \approx \sum_{n=-\infty}^{+\infty} (u_n(t)e^{nik_m x} + u_n^*(t)e^{-nik_m x}) \tag{12}$$

where $u_n(t)$ is the complex amplitude of the $n^{th}$ harmonics. Let the amplitude corresponding to the leading harmonics ($n = 1$) is $\phi(t)$. After assuming that the amplitude of every other harmonics can be written as a power series of $\phi(t)$, substituting *Equation 12* into *Equation 9* allows to write an equation that describes the evolution of $\phi(t)$. In the case when the resulting equation is of Stuart-Landau type:

$$\frac{d\phi}{dt} = c_1\phi + c_2\phi^3 - c_3\phi^5 \tag{13}$$

with $c_1, c_2, c_3 > 0$, this corresponds to the normal form of a sub-critical pitchfork bifurcation (*Strogatz, 2018*). Together with the condition given by *Equation 11*, the existence of a sub-critical PB for the full system (*Equation 9*) is guaranteed. A numerical or analytical analysis of *Equation 13* enables the identification of the position of the $SN_{PB}$.

## Modeling EGFR phosphorylation polarization dynamics

The dynamics of the experimentally identified spatially distributed EGFR sensing network (*Figure 1B*, *Figure 1—figure supplement 1B*) is described using the following one-dimensional system of partial differential equations (PDEs):

$$\frac{\partial [E_p]}{\partial t} = f_1([E_p], [E - E_p], [RG_a], [N2_a], [EGF_t]) + D_{E_p} \frac{\partial^2 [E_p]}{\partial x^2}$$

$$\frac{\partial [E - E_p]}{\partial t} = f_2([E_p], [E - E_p], [EGF_t]) + D_{E-E_p} \frac{\partial^2 [E - E_p]}{\partial x^2}$$

(14)

$$\frac{\partial [RG_a]}{\partial t} = f_3([E_p], [E - E_p], [RG_a]) + D_{RG_a} \frac{\partial^2 [RG_a]}{\partial x^2}$$

$$\frac{\partial [N2_a]}{\partial t} = f_4([E_p], [E - E_p], [N2_a])$$

with

$$f_1 = ([E_t] - [E_p] - [E - E_p])(\alpha_1([E_t] - [E_p] - [E - E_p]) + \alpha_2[E_p] + \alpha_3[E - E_p]) -$$

$$\gamma_1[RG_a][E_p] - \gamma_2[N2_a][E_p] - k_{on}([EGF_t] - [E - E_p])[E_p]^2 + 1/2k_{off}[EE_p];$$

$$f_2 = k_{on}([EGF_t] - [E - E_p])([E_p]^2 + ([E_t] - [E_p] - [E - E_p])^2) - k_{off}[E - E_p];$$

$$f_3 = k_1([RG_t] - [RG_a]) - k_2[RG_a] - \beta_1[RG_a]([E_p] + [E - E_p]);$$

and

$$f_4 = \epsilon(k_1([N2_t] - [N2_a]) - k_2[N2_a] + \beta_2([E_p] + [E - E_p])([N2_t] - [N2_a])).$$

The reaction terms are described in details in *Stanoev et al., 2018*. In brief, $[E - E_p]$ is the phosphorylated ligand-bound dimeric EGFR, $[E_p]$ - ligandless phosphorylated EGFR, $[E_t]$ - total amount of EGFR, $[RG_a], [RG_t]$ and $[N2_a], [N2_t]$ - the active and total amount of the membrane localized PTPRG and the ER-bound PTPN2, respectively. Both, the receptor and the deactivating enzymes have active and inactive states, and the model equations describe their state transition rates. Therefore, mass is conserved in the system and the total protein concentrations of the three species ($[E_t]$, $[RG_t]$ and $[N2_t]$) are constant parameters. Autonomous, autocatalytic and ligand-bound-induced activation of ligandless EGFR ensue from bimolecular interactions with distinct rate constants $\alpha_{1-3}$, respectively. Other parameters are as follows: $k_1/k_2$ — activation/inactivation rate constants of the phosphatases, $\beta_1/\beta_2$ - receptor-induced regulation rate constants of *PTPRG/PTPN*2, $\gamma_1/\gamma_2$ - specific reactivity of the enzymes (*PTPRG/PTPN*2) towards the receptor. The EGFR-PTPN2 negative feedback is on a time scale ($\epsilon$) approximately two orders of magnitude slower than the phosphorylation-dephosphorylation reaction, as estimated from the ~ 4 min recycling time of $EGFR_p$ (*Stanoev et al., 2018*). This enables, when necessary, to consider a quasi-steady state approximation for the dynamics of PTPN2 for simplicity:

$$[N2_a]_{qss} = [N2_t] \cdot \frac{(k_1 + \beta_2 \cdot ([E_p] + [E - E_p]))}{k_1 + k_2 + \beta_2 \cdot ([E_p] + [E - E_p])}$$

(15)

$[EGF_t]$ denotes the total ligand concentration. Assuming that at low, physiologically relevant EGF doses, the ligand will be depleted from the solution due to binding to EGFR (*Lauffenburger and Linderman, 1996*), ligand-binding unbinding was explicitly modeled ($k_{on}$, $k_{off}$) in *Equation 14*, with values corresponding to the experimentally identified ones.

The diffusion terms model the lateral diffusion of the EGFR and PTPRG molecules on the plasma membrane, whereas PTPN2 is ER-bound and does not diffuse. Single particle tracking studies have demonstrated that EGFR molecules on the plasma membrane occupy three distinct mobility states, free, confined and immobile, with the occupations of the free and immobile states decreasing and increasing significantly after EGF stimulation, respectively (2 min after EGF stimulation, corresponding with the time-scale of EGF binding) (*Ibach et al., 2015*). In the reaction-diffusion (RD) simulations therefore for simplicity, it is assumed that $D_{E-E_p} \approx 0$, whereas diffusion constants of same order are assumed for the ligandless EGFR and PTPRG ($D_{E_p} \sim D_{RG_a}$).

## Analytical consideration for an $SN_{PB}$ existence in the EGFR network

To identify analytically the existence of a $SN_{PB}$ in the EGFR receptor network, we performed a weakly nonlinear analysis as described in the general consideration (Section. Theoretical consideration of the navigation mechanism in a generalized reaction-diffusion signaling model). For this, we considered the system *Equation 14*, where the dynamics of PTPN2 is at quasi-steady state (*Equation 15*), $[E - E_p] = 0$, and rest of the dependent and independent variables were scaled to have a dimensionless form. Let $[\tilde{E}_p] = [E_p]/E_0$, $[\tilde{RG}_a] = [RG_a]/RG_0$, $\tilde{x} = x/x_0$, $\tau = t/t_0$, such that $t_0 = 1/(k_1 + k_2)$, $E_0 = k_1/\beta_2$, $RG_0 = (k_1 + k_2)/\gamma_1$ and $t_0/x_0^2 = 1/D_{E_p}$. Substituting these into *Equation 14* yields the system of dimensionless equations:

$$\frac{\partial [\tilde{E}_p]}{\partial \tau} = q_1 + q_2[\tilde{E}_p] + q_3[\tilde{E}_p]^2 - [\tilde{RG}_a][\tilde{E}_p] - \frac{q_4(1 + [\tilde{E}_p])[\tilde{E}_p]}{(1 + k + [\tilde{E}_p])} + \frac{\partial^2 [\tilde{E}_p]}{\partial \tilde{x}^2}$$

$$\frac{\partial [\tilde{RG}_a]}{\partial \tau} = r_1 - [\tilde{RG}_a] - r_2[\tilde{RG}_a][\tilde{E}_p] + D\frac{\partial^2 [\tilde{RG}_a]}{\partial \tilde{x}^2}$$

(16)

with $q_1 = \frac{\alpha_1 \cdot [E_t]^2}{(k_1+k_2)\cdot\beta_2}$, $q_2 = \frac{(\alpha_2 - 2\cdot\alpha_1)\cdot[E_t]}{k_1+k_2}$, $q_3 = \frac{(\alpha_1 - \alpha_2)\cdot k_t}{(k_1+k_2)\cdot\beta_2}$, $q_4 = \frac{\gamma_2\cdot[N2_t]}{k_1+k_2}$, $k = k_2/k_1$, $r_1 = \frac{k_1\cdot[RG_t]\cdot\gamma_1}{(k_1+k_2)^2}$, $r_2 = \frac{\beta_1\cdot k_1}{(k_1+k_2)\cdot\beta_2}$ and $D = \frac{D_{RG_\alpha}}{D_{E_p}}$.

We further simplify the system *Equation 16* by taking the Talyor series expansion of the quasi-steady state approximation of $[N2_a]$ around $E_s$, the steady state of $[\tilde{E}_p]$

$$\frac{q_4(1 + [\tilde{E}_p])[\tilde{E}_p]}{1 + k + [\tilde{E}_p]} = q_7 + q_8[\tilde{E}_p] + q_9[\tilde{E}_p]^2 + o([\tilde{E}_p]^2)$$

(17)

with $q_7 = \frac{E_s q_4}{1+k+E_s} - \frac{E_s q_4(1+k)}{(1+k+E_s)^2}$, $q_8 = \frac{E_s q_4}{1+k+E_s} + \frac{q_4(1+k)}{(1+k+E_s)^2}(1 - E_s)$, and $q_9 = \frac{q_4(1+k)}{(1+k+E_s)^2}$, thus yielding:

$$\frac{\partial [\tilde{E}_p]}{\partial \tau} = q_9 + q_{10}[\tilde{E}_p] + q_{11}[\tilde{E}_p]^2 - [\tilde{RG}_a][\tilde{E}_p] + \frac{\partial^2 [\tilde{E}_p]}{\partial \tilde{x}^2}$$
$$\frac{\partial [\tilde{RG}_a]}{\partial \tau} = r_1 - [\tilde{RG}_a] - r_2[\tilde{RG}_a][\tilde{E}_p] + D\frac{\partial^2 [\tilde{RG}_a]}{\partial \tilde{x}^2}$$

(18)

with $q_9 = q_1 - q_7$, $q_{10} = q_2 - q_8$ and $q_{11} = q_3 - q_9$.

To avoid long expression in the further analysis, we re-name the dependent variables as $u_1 = [\tilde{E}_p]$ and $u_2 = [\tilde{RG}_a]$, and the independent variables as $\tilde{x} = x$, $\tau = t$. The system *Equation 16* therefore obtains the generic form:

$$\frac{\partial u_1}{\partial t} = F_1(u_1, u_2) + \frac{\partial^2 u_1}{\partial x^2}$$

$$\frac{\partial u_2}{\partial t} = F_2(u_1, u_2) + D\frac{\partial^2 u_2}{\partial x^2}.$$

(19)

In order to perform linear stability analysis, a one-dimensional projection of *Equation 19* is considered,

$$\frac{du_{1f}}{dt} = F_1(u_{1f}, u_{2f}) - (u_{1f} - u_{1b}) = G_1(u_{1f}, u_{2f}, u_{1b})$$

$$\frac{du_{2f}}{dt} = F_2(u_{1f}, u_{2f}) - D(u_{2f} - u_{2b}) = G_2(u_{1f}, u_{2f}, u_{2b})$$

(20)

$$\frac{du_{1b}}{dt} = F_1(u_{1b}, u_{2b}) - (u_{1b} - u_{1f}) = G_3(u_{1b}, u_{2b}, u_{1f})$$

$$\frac{du_{2b}}{dt} = F_2(u_{1b}, u_{2b}) - D(u_{2b} - u_{2f}) = G_4(u_{1b}, u_{2b}, u_{2f})$$

The simplified one-dimensional geometry assumes a model composed of two compartments (front and back), resembling a projection of the membrane along the main diagonal of the cell. The

standard approach of modeling the diffusion along the membrane in this case is simple exchange of the diffusing components. The one-dimensional projection, as demonstrated below, preserves all of the main features of the PDE model.

Let, $U_s = \begin{pmatrix} u_{1fs} \\ u_{2fs} \\ u_{1bs} \\ u_{2bs} \end{pmatrix}$ be the stable symmetric steady state of the system ($u_{1fs} = u_{1bs}$, $u_{2fs} = u_{2bs}$). A small

amplitude perturbation on this symmetric steady state of the form,

$$
\begin{pmatrix} u_{1f}(t) \\ u_{2f}(t) \\ u_{1b}(t) \\ u_{2b}(t) \end{pmatrix} = \begin{pmatrix} u_{1fs} \\ u_{2fs} \\ u_{1bs} \\ u_{2bs} \end{pmatrix} + \begin{pmatrix} \delta u_{1f} \\ \delta u_{2f} \\ \delta u_{1b} \\ \delta u_{2b} \end{pmatrix} \cdot e^{\lambda t}
\tag{21}
$$

yields a linearized equation,

$$
\lambda \begin{pmatrix} \frac{d\delta u_{1f}}{dt} \\ \frac{d\delta u_{2f}}{dt} \\ \frac{d\delta u_{1b}}{dt} \\ \frac{d\delta u_{2b}}{dt} \end{pmatrix} = \mathbf{J} \begin{pmatrix} \delta u_{1f} \\ \delta u_{2f} \\ \delta u_{1b} \\ \delta u_{2b} \end{pmatrix}
\tag{22}
$$

where

$$
\mathbf{J} = \begin{pmatrix}
\frac{\partial G_1}{\partial u_{1f}} & \frac{\partial G_1}{\partial u_{2f}} & \frac{\partial G_1}{\partial u_{1b}} & 0 \\
\frac{\partial G_2}{\partial u_{1f}} & \frac{\partial G_2}{\partial u_{2f}} & 0 & \frac{\partial G_2}{\partial u_{2b}} \\
\frac{\partial G_3}{\partial u_{1f}} & 0 & \frac{\partial G_3}{\partial u_{1b}} & \frac{\partial G_3}{\partial u_{2b}} \\
0 & \frac{\partial G_4}{\partial u_{2f}} & \frac{\partial G_4}{\partial u_{1b}} & \frac{\partial G_4}{\partial u_{2b}}
\end{pmatrix}
$$

is the Jacobian of the system evaluated at the symmetric steady state. In order to identify existence of $PB$ in the system, the condition given in *Equation 11* should be satisfied for an odd mode of the perturbation. For the one-dimensional projection (*Equation 20*), the odd mode of the perturbation ($\delta \mathbf{U}(-\mathbf{x}) = -\delta \mathbf{U}(\mathbf{x})$) must yield: $\delta u_{1f} = -\delta u_{1b}$ and $\delta u_{2f} = -\delta u_{2b}$. Substituting this into *Equation 22* to obtain $F_-(\lambda)$, in the limit $\lambda \to 0$ renders:

$$
\lim_{\lambda \to 0} F_-(\lambda) = det \begin{pmatrix}
(\frac{\partial G_1}{\partial u_{1f}} + \frac{\partial G_3}{\partial u_{1b}}) - (\frac{\partial G_1}{\partial u_{1b}} + \frac{\partial G_3}{\partial u_{1f}}) & (\frac{\partial G_1}{\partial u_{2f}} + \frac{\partial G_2}{\partial u_{2b}}) \\
(\frac{\partial G_2}{\partial u_{1f}} + \frac{\partial G_4}{\partial u_{1b}}) & (\frac{\partial G_2}{\partial u_{2f}} + \frac{\partial G_4}{\partial u_{2b}}) - (\frac{\partial G_2}{\partial u_{2b}} + \frac{\partial G_4}{\partial u_{2f}})
\end{pmatrix} = 0
\tag{23}
$$

Thus, there exists parameter set for which existence of PB in the system *Equation 20* is guaranteed.

To identify whether the PB is sub-critical and thereby identify existence of a $SN_{PB}$, the solution of the system *Equation 19* is approximated as in *Equation 12*:

$$
u(x,t) = \phi(t)e^{ik_m x} + \phi^*(t)e^{-ik_m x} + u_0(t) + \sum_{n=2}^{3} (u_n(t)e^{nik_m x} + u_n^*(t)e^{-nik_m x})
$$

$$
v(x,t) = \phi(t)e^{ik_m x} + \phi^*(t)e^{-ik_m x} + v_0(t) + \sum_{n=2}^{3} (v_n(t)e^{nik_m x} + v_n^*(t)e^{-nik_m x})
$$

$$
\tag{24}
$$

The expansion is taken to $n = 3^{rd}$ order, rendering an amplitude equation of $5^{th}$ order. As described in *Becherer et al., 2009*, the complex coefficients of the $n = 0^{th}$, $n = 2^{nd}$ and $n = 3^{rd}$ harmonics can be approximated as power series of $\phi(t)$. Substituting into *Equation 19* allows to derive these coefficients. This yields a system of coupled ODEs representing the time evolution of the complex amplitudes, in this case, for $\phi(t)$, $u_0(t)$, $v_0(t)$, $u_1(t)$, $v_1(t)$, $u_2(t)$, $v_2(t)$, $u_3(t)$ and $v_3(t)$. Assuming that the dynamics of the higher order harmonics reaches their steady state much faster than the leading perturbation does, the derivatives of their amplitudes can be set to zero. This allows to obtain expressions of the amplitudes purely as functions $\phi$ and the parameters of the system as:

$$u_0(\phi) = (\frac{1}{q_{10}}(2(1 - q_{11}) - \frac{q_9}{|\phi|^2}))|\phi|^2$$

$$v_0(\phi) = (\frac{r_1}{|\phi|^2} - 2r_2)|\phi|^2$$

$$u_2(\phi) = u_2^{(2)}\phi^2$$

$$v_2(\phi) = v_2^{(2)}\phi^2 \qquad (25)$$

$$u_3(\phi) = u_3^{(3)}\phi^3$$

$$v_3(\phi) = v_3^{(3)}\phi^3$$

where $u_2^{(2)} = \frac{1-q_{11}}{q_{10}-4k_m^2}$, $v_2^{(2)} = \frac{-r_2}{1+4Dk_m^2}$, $u_3^{(3)} = \frac{u_2^{(2)}+v_2^{(2)}-2q_{11}u_2^{(2)}}{q_{10}-9k_m^2}$ and $v_3^{(3)} = \frac{-r_2(u_2^{(2)}+v_2^{(2)})}{1+9Dk_m^2}$. The dynamics of the leading harmonics ($n = 1$) can be written as:

$$\frac{d\phi}{dt} = c_1\phi + c_2\phi^3 - c_3\phi^5 \qquad (26)$$

where $c_1 = q_{10} - k_m^2 - r_1 + \frac{q_9(1-2q_{11})}{q_{10}}$, $c_2 = (1 - q_{11})(2q_{11} - 1)(\frac{2}{q_{10}} - \frac{1}{q_{10}-4k_m^2}) + r_2(2 + \frac{1}{1+4Dk_m^2})$ and $c_3 = 2q_{11}u_2^{(2)}u_3^{(3)} - u_2^{(2)}v_3^{(3)}$. *Equation 26* is of Stuart-Landau type and represents a normal form of a sub-critical pitchfork bifurcation. This shows the existence of $SN_{PB}$ in the EGFR network.

To corroborate this, we also performed numerical bifurcation analysis on one-dimensional projection (*Equation 20*) where the reaction terms have the form as defined in *Equation 14*, including the full form for $[N2_a]$, when $[E - E_p] = 0$. The bifurcation analysis (*Figure 1—figure supplement 1C*) was obtained using the Xppaut software package (*Ermentrout, 2016*). The parameters in the model *Equation 14* have been described in *Stanoev et al., 2018*, where they were calibrated with experimental data: $\alpha_1 = 0.001$, $\alpha_2 = 0.3$, $\alpha_3 = 0.7$, $\beta_1 = 11$, $\beta_2 = 1.1$, $k_1 = 0.5$, $k_2 = 0.5$, $g_1 = 1.9$, $g_2 = 0.1$, $k_{on} = 0.05$, $k_{off} = 0.28$, $\epsilon = 0.01$, $RG_t = 1$, $N2_t = 1$; and the diffusion-like terms have been scaled from the values derived in *Orr et al., 2005*: $\tilde{D}_{E_p} = 0.02$, $\tilde{D}_{RG_a} = 0.02$ (see also *Supplementary file 1*).

The bifurcation analysis is performed with respect to total EGFR concentration at the plasma membrane in order to reveal all possible dynamical regimes of the system. This analysis demonstrates that for the spatially distributed EGFR network, the homogeneous steady state (HSS, gray solid line, *Figure 1—figure supplement 1C*) representing basal non-polarized state losses stability via a symmetry-breaking pitchfork bifurcation (*PB*), which gives rise to a polarized state represented via an inhomogeneous steady states (IHSS). The polarized state is stabilised via saddle-node bifurcations ($SN_{PB}$) (*Figure 1—figure supplement 1C*, magenta branched lines). There is a coexistence between the HSS and the IHSS before the *PB*, rendering it sub-critical. The IHSS (*Koseska et al., 2013*) that gives rise to the stable polarized state is a single attractor that describes a heterogeneous state with two branches corresponding to orientation of the front-back-polarized state. The IHSS solution is therefore fundamentally distinct from a bistable system where the high and the low phosphorylation states correspond to two different homogeneous steady states. As the IHSS is a single attractor, the high and low phopshorylation state are interdependent, rendering the *PB* a unique mechanism for generating robust front-back polarization.

We next describe the dynamical basis of the polarization and memory of polarization in details. We assume that the steady state EGFR concentration at the plasma membrane corresponds to

organization at criticality, before the $SN_{PB}$. For this receptor concentration, only the basal unpolarized state (HSS) is stable (**Figure 1—figure supplement 1A**, top left, schematic representation). In the presence of a spatially inhomogeneous EGF signal however, the system undergoes a series of complex transitions through which the topology of the phase space changes. In particular, the inhomogeneity introduced by the localized signal leads to unfolding of the pitchfork bifurcation, such that for the same organization (the given EGFR concentration), only the polarized state (the IHSS) is stable (**Figure 1—figure supplement 1A**, top right). This unfolding of the $PB$ therefore enables robust transition from basal to polarized state. When the EGF signal is removed, the system undergoes again topological phase space changes. However, in this transition, the system does not revert back to the unpolarized state immediately, but rather it is transiently maintained in the "ghost" of the $SN_{PB}$ that is lost in this transition (**Figure 1—figure supplement 1A**, low). This is manifested as a transient memory of the polarized state, after which the system rapidly reverts to the basal state.

The reaction diffusion simulations were performed by assuming PTPN2 at quasi-steady state. The cell boundary was represented with a 1D circular domain of length $L = 2\pi R$ (where $R = 2\mu m$) which was then divided into 20 equal bins. The diffusion terms were approximated by central difference method, enabling for conversion of the PDE system to a system of ordinary differential equations (ODEs). Stochastic simulations with additive white noise were implemented by adding $\sigma \cdot dW_t$ ($\sigma = 0.02$, $dW_t$ is sampled from a normal distribution with mean 0 and variance 0.01) in the equation for $[E_p]$. The stochastic *sdeint* Python package was used. Parameters: $D_{E_p} = D_{RG_a} = 0.008\ \mu m^2/min$. $D_{E_p}$ was taken from **Orr et al., 2005** and scaled to correspond to a cell with perimeter $L$ in the simulations. For organization in the homogenous symmetric steady states (the basal and pre-activated states), organization at criticality or in the stable polarized state (IHSS), $E_t \in \{1.1, 1.85, 1.26, 1.35\}$ respectively, time step was set to 0.01 min, other parameters as above. Periodic boundary conditions were used. To mimic the dynamic nature of $EGF^{647}$ gradient, a Gaussian function on a periodic window with varying amplitude and standard deviation was used (shape shown in **Figure 1D**, top). To represent the state-space trajectory (**Figure 1F**, bottom), stochastic realization of the one-dimensional projection of the full system (as for the bifurcation analysis) was used.

## Physical model of single-cell chemotaxis

To describe signal-induced cell shape changes and subsequent cell migration, we combined the dynamical description of the gradient sensing capability of the EGFR network (**Equation 14**, **Figure 1B**) together with a physical model for cellular migration, thereby implicitly modeling the signal-induced cell shape changes (**Figure 1C**). In order to couple a mechanical model of the cell with the biochemical EGFR signaling model as a means to simulate large cellular deformations, we utilized the Level Set Method (LSM) (**Osher and Sethian, 1988**) as described in **Yang et al., 2008**. Briefly, the cell boundary at time $t$ is described on a two-dimensional Cartesian grid by the closed-contour $\Gamma(t) = \{\mathbf{x}|\Psi(\mathbf{x}, t) = 0\}$, that represent the zero-level set of the potential function $\Psi(\mathbf{x}, t)$, taken to have an initial form:

$$\Psi(\mathbf{x}, 0) = \begin{cases} -d(\mathbf{x}, \Gamma), & \text{if } \mathbf{x} \in S \\ d(\mathbf{x}, \Gamma), & \text{if } \mathbf{x} \notin S \\ 0, & \text{if } \mathbf{x} \in \Gamma \end{cases} \tag{27}$$

where $S$ identifies the area occupied by the cell and $d(\mathbf{x}, \mathbf{\Gamma})$ is the distance of position $x$ to the curve $\Gamma$. Thus, the cell membrane is represented implicitly through the potential function which is defined on the fixed Cartesian grid, eliminating the need to parameterize the boundary, and thereby enabling to handle complex cell boundary geometries.

The shape of the cell ($\Gamma(\mathbf{x}, t)$) evolves according to the Hamilton-Jacobi equation:

$$\frac{\partial \Psi(\mathbf{x}, t)}{\partial t} + \mathbf{v}(\mathbf{x}, t) \cdot \nabla \Psi(\mathbf{x}, t) = 0 \tag{28}$$

The vector $\mathbf{v}(\mathbf{x}, t)$ is the velocity of the level set moving in the outward direction, thereby intrinsically describing the cell's membrane protrusion and retraction velocities that are driven by internally generated mechanical forces (e.g. actin polymerization or myosin-II retraction, **Bray, 2000**). To determine how these forces translate to membrane velocity, a mechanical model that describes the viscoelastic behavior of the cell represented as a viscoelastic cortex surrounding a viscous core, is implemented.

Following *Yang et al., 2008*, the cortex connecting the cell membrane and the cytoplasm is represented by a Voigt model parallel connection of an elastic element $k_c$ and a viscous element $\tau_c$, whereas the cytoplasm is modeled as a purely viscous element, $\tau_a$, which is placed in series with the Voigt model.

Let $\mathbf{l}(\mathbf{x}, t)$, $\mathbf{x} \in \Gamma(t)$ be the viscoelastic state of the cell at time $t$ and at a position $\mathbf{x}$ on the membrane, such that $\|\mathbf{l}\|$ represents the length of the numerous parallel unconnected spring-damper systems. The viscoelastic state of the cell then evolves according to:

$$\frac{-k_c}{\tau_c}\mathbf{l}(t) + \frac{1}{\tau_c}\mathbf{P_{total}}(t) = \nabla\mathbf{l} \cdot \mathbf{v(t)} + \frac{\partial\mathbf{l}(t)}{\partial t} \tag{29}$$

where $\nabla$ is the gradient operator, the pressure $\mathbf{P_{total}}(t) = \mathbf{P_{prot}}(t) + \mathbf{P_{retr}}(t) + \mathbf{P_{area}}(t) - \mathbf{P_{ten}}(t)$ is sum of the protrusion, retraction, area conservation, and cortical tension pressures, respectively. The EGFR signaling state ($[E_p]$) directly determines the protrusion/retraction pressure, since high/low signaling activity triggers actin polymerization / myosin-II retraction following: $\mathbf{P_{prot}}(t) = K_{prot}(([E_p](t) - <[E_p](t)>)/([E_p]_{max}(t) - <[E_P](t)>))\mathbf{n}$ and $\mathbf{P_{ret}}(t) = -K_{retr}((<[E_p]> - [E_p])/(<[E_P]> - [E_p]_{max}))\mathbf{n}$, where $<.>$ denotes mean at the membrane, $K_{prot}$, $K_{retr}$ - proportionality constants. The cell is assumed to be flat with uniform thickness, such that the 2D area ($A(t)$) of the cell is conserved ($\mathbf{P_{area}(t)} = K_{area}(A(0) - A(t))\mathbf{n}$), $K_{area}$ - proportionality constant. The pressure generated by the cortical tension therefore depends only on the 2D local surface curvature and the 2D equilibrium pressure, rendering the rounding pressure due to cortical tension to be $\mathbf{P_{ten}}(t) = K_{ten}(\kappa(\Gamma) - 1/R)\mathbf{n}$, with $\kappa(x)$ being the local membrane curvature, R - initial cell radius, was set to 2 $\mu m$, and $K_{ten}$ - proportionality constant. The local membrane velocity $\mathbf{v(x)}$, $\mathbf{x} \in \Gamma(t)$ depends both on the viscoelastic nature of the cell and on the effective pressure profile ($\mathbf{P_{total}}(t)$) and is given by,

$$\mathbf{v} = \frac{-k_c}{\tau_c}\mathbf{l} + (\frac{1}{\tau_c} + \frac{1}{\tau_a})\mathbf{P_{total}} \tag{30}$$

For the simulations in *Figures 1 and 4* and *Figure 4—figure supplements 1 and 2* first the stochastic PDEs (*Equation 14*) are solved and the kymographs of the signalling ($[E_P]$) activity are generated. The viscoelastic state is initialized with zero value on the membrane, $\mathbf{l}(\mathbf{x}, 0) = 0$. At each time point, $\mathbf{P_{total}}$ is estimated, as well as the local membrane velocity using *Equation 30*. This velocity is then used to evolve both the viscoelastic state (*Equation 29*) and the potential function (*Equation 27*).

The spatial discretization of these advection equations (*Equations 28 and 29*) was performed using the *upwindENO2* scheme, as described in the Level Set Toolbox (*Mitchell, 2007*) and was integrated with first-order forward Euler method. The time step was set to $0.01 min$ and the potential function was solved on a 2D Cartesian grid with spatial discretization of 5 points per μm. All the codes were custom implemented in Python. Parameters: $k_c = 0.1\,nN/\mu m^3$, $\tau_c = 0.08\,nN\,min/\mu m^3$, $\tau_a = 0.1\,nN\,min/\mu m^3$, $K_{prot} = 0.08\,nN/\mu m^2$, $K_{retr} = 0.05\,nN/\mu m^2$, $K_{area} = 0.02\,nN/\mu m^4$, $K_{ten} = 0.1\,nN/\mu m$. $K_{ten}$ was taken from the literature, corresponding to an experimentally measured range of cell cortical tension values (*Cartagena-Rivera et al., 2016*). The rest of the parameters were selected to match the cell migration speed during gradient and memory phase, estimated from the experiments (*Figure 3A*, $v = 0.49 \pm 0.173\mu m/min$).

## Acknowledgements

The authors thank Angel Stanoev for initial analysis of the EGFR polarity model, critical discussion during the project, as well as valuable comments on the manuscript, Manish Yadav and Monika Scholz for critical reading of the manuscript and valuable suggestions, and Frédéric Paquin-Lefebvre for valuable suggestions on the realization of the reaction-diffusion simulations. All of the experiments were carried in the lab of Philippe Bastiaens, and we are particularly grateful for the opportunity to be part of that engaging and critical community where we could learn and develop this project. We especially thank P Bastiaens for numerous critical discussion and suggestions that were crucial throughout the project, as well as for detailed comments that significantly helped us to improve the manuscript. Data and code availability: All data generated or analysed during this study is included in the manuscript and supporting files. The codes for the numerical simulations are available under https://github.

com/akhileshpnn/Cell-memory. Funding: The project was funded by the Max Planck Society, partially through the Lise Meitner Excellence Program.

## Additional information

### Funding

| Funder | Grant reference number | Author |
|--------|------------------------|--------|
| Max Planck Society | | Aneta Koseska |

The funders had no role in study design, data collection and interpretation, or the decision to submit the work for publication.

### Author contributions

Akhilesh Nandan, Data curation, Formal analysis, Investigation, Methodology, Software, Validation, Visualization, Writing – review and editing; Abhishek Das, Data curation, Formal analysis, Investigation, Software, Validation, Visualization, Writing – review and editing; Robert Lott, Formal analysis, Investigation, Software, Validation, Visualization, Writing – review and editing; Aneta Koseska, Conceptualization, Formal analysis, Funding acquisition, Investigation, Methodology, Project administration, Supervision, Validation, Writing - original draft, Writing – review and editing

### Author ORCIDs

Aneta Koseska ⓘ http://orcid.org/0000-0003-4263-2340

### Decision letter and Author response

Decision letter https://doi.org/10.7554/eLife.76825.sa1
Author response https://doi.org/10.7554/eLife.76825.sa2

## Additional files

### Supplementary files

- Supplementary file 1. Model parameters. Details included also in Methods.
- Transparent reporting form

### Data availability

Source data is provided with the submission. The numerical data used to generate the corresponding figures can be obtained from the codes deposited in https://github.com/akhileshpnn/Cell-memory, (copy archived at swh:1:rev:288921244e5042922e1bbddcf5037a5e87e78723).

The following dataset was generated:

| Author(s) | Year | Dataset title | Dataset URL | Database and Identifier |
|-----------|------|---------------|-------------|--------------------------|
| Akhilesh N | 2022 | Cells use molecular working memory to navigate in changing chemoattractant fields | https://github.com/akhileshpnn/Cell-memory | GitHub, Cell-memory |

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
