## [Editor Report]

This paper addresses how cells can robustly maintain direction during movement by ignoring noise in concentration gradients while also being able to adapt to new signals in those gradients. The authors study this tension in EGFR signaling by postulating a form of cellular memory in a theoretical framework based on dynamical systems and bifurcation theory. The authors also carry out experiments that raise further interesting questions. This paper will be of interest to scientists of all stripes working on cell motility and for theorists who take a dynamical systems view of biological phenomena.

---

## [Decision Letter]

**Decision letter after peer review:**

Thank you for submitting your article "Cells use molecular working memory to navigate in changing chemoattractant fields" for consideration by *eLife*. Your article has been reviewed by 3 peer reviewers, and the evaluation has been overseen by a Reviewing Editor and Naama Barkai as the Senior Editor. The following individual involved in the review of your submission has agreed to reveal their identity: Elizabeth R Jerison (Reviewer #3).

Essential revisions:

The reviewers broadly appreciate the theoretical framework here and the new experimental data gathered. However, a few critical changes are required before publication.

1) The most critical change suggested by the reviewers is tempering and clarifying the relationship between experimental results and theory here. The theoretical work here is already sufficiently interesting, and the experimental results add to it, but the current statements imply a consistency between theory and experiments that is not fully supported.

The authors should temper their claims and clearly delineate the ways in which the experimental results agree and disagree with the theoretical model here. (`All models are wrong, but some are useful.') Such a clear statement would allow future work to build on the results here.

2) The authors should discuss other predictions of their model for future experiments (e.g., see reviewer #2's and #3's comments).

3) Simplify/reduce technical jargon – e.g., see reviewer #1's comments.

*Reviewer #1 (Recommendations for the authors):*

Appraisal of results supporting Figure #1 and theoretical background:

The theoretical background is sound and detailed. However, much of it rests on the choice of parameters as identified by the authors. It would be beneficial to see the dependence of the existence of saddle-node pitchfork bifurcations (and similarly the stable polarisation state, and homogeneous symmetric steady states) on key parameters (especially the total EGFR on the plasma membrane, and the total amount of the membrane-localized PTPRG and the ER-bound PTPN2).

Also, the authors claim that in their model, "The cell polarity is sustained even when the EGF signal is briefly disrupted, but also, the cell is able to rapidly reverse the direction of polarization when the signal direction is inverted". It would be helpful if the authors indicated (either by a numerical plot, or analytical calculations if possible) the time scales of such responses in their model. Chiefly, it would help to know how long the signal would need to be disrupted before the polarity is no longer sustained, how long the signal direction needs to be inverted before the cell reverses polarity, and how long it takes for the cell to either reverse or lose polarity.

Appraisal of results supporting conclusion #1:

The authors claim that MCF7 cells maintain a memory of the direction of previously encountered signals through prolonged EGFR phosphorylation polarisation and temporal evolution of the cell protrusion that emerges from a saddle-node pitchfork bifurcation which maintains the system away from the steady states. The key pieces of evidence present claim that after subjecting the cells to a stable gradient of EGF for one hour, the cells remain polarised for ~40 minutes. While the experimental evidence shows that the cells are partially polarised for ~40 minutes after gradient washout, the polarisation is less than it is in the presence of the signal and steadily approaches the unpolarised state (this is most clearly seen in the single-cell trajectories supplied in Supp. Figure 2-1F). This is unlike the distinct persistence of polarisation shown in the numerical results (Figure 1F and G). To me, the experimental data suggest that the timescale of relaxation from the polarized state to the unpolarized state due to slow dephosphorylation kinetics is ~40 minutes. The authors should describe what the characteristic chemical timescales of the system are, and if persistence of ~40 minutes is uncharacteristically large. How do the experimental results reflect the existence of a metastable "ghost" state as in saddle-node bifurcations, rather than two steady states that take ~40 minutes to move between them as in a subcritical pitchfork bifurcation without a saddle-node? The authors attempt to answer this by using Lapatinib later on (which should be brought into this section), but the difference due to Lapatinib isn't established as data is shown only for one cell, and further, the unintended effects of Lapatinib on the cells hasn't been shown (a possible way to demonstrate this would be a control experiment with cells treated with Lapatinib for a short amount of time and then exposed to a gradient of EGF to show that there is a memory in such a case).

Appraisal of results supporting conclusion #2:

The authors claim that the transient memory encoded in the EGFR phosphorylation polarisation and cell protrusion translates to memory in directional migration up a linear EGF gradient that persisted after the gradient washout. The authors demonstrate that there is a transient memory in directional migration after subjecting the cells to 5 hours of a linear gradient (however, they do not initially identify the period of this memory in Figure 3A and 3B. They later suggest that it is 50 minutes after gradient washout); however, it is difficult to see if the memory encoded in the EGFR phosphorylation polarisation and cell protrusion is accurately reflected in the memory in directional migration due to the vast differences in the timescales of the experimental evidence for the two (the former is in a constant exposure for 1 hour, while the latter is for a constantly decreasing exposure for 5 hours). Further, the evidence using Lapatinib is not conclusive, as even if we believe that there is a difference between the directionality with and without Lapatinib (Figure 3J), this does not establish the link between transient molecular states and transient directionality. The evidence in Figure 3G is inconclusive as it is not sufficiently different from the case without Lapatinib and Takens's delay-embedding trajectory is based on just one cell.

A key piece whose repercussions were not discussed much by the authors is that the gradient steepness was progressively decreased. The authors treated the entire period before the gradient washout as being the same, and I am still unclear as to what the effect of decreasing the gradient was.

Appraisal of results supporting conclusion #3:

The authors claim that MCF10A cells can sense dynamic gradients while maintaining robust directional migration even when the signal was disrupted. This is seen very clearly in silico in Figure 4B, but the experimental results in Figure 4C clearly differ in that the cells do not seem to have any significant memory following the gradient washout. Further, the ability of the cells to respond to changes in the environment should be contrasted to the case of slow dephosphorylation kinetics to highlight the role of the transient memory state. In general, it would be helpful to understand what the effect of a shorter time scale of the transition from a polarised to unpolarised state would be on the robustness of the system, and what the effect of a longer time scale would be on the adaptation of the system.

Note regarding the Discussion:

The authors do not distinguish their contributions from those made by other research groups that are addressing similar questions. For example, in line 310, the authors cited a set of papers as relevant experimentally but dismissed them as not providing a mechanism even though some of them did. For instance, Skoge et al. (PNAS, 2014), discussed a possible mechanism by which cells exhibit memory and maintain directed motion in the classical back-of-the-wave problem for *Dictyostelium*, and Prentice-Mott et al. did so for chemotactic neutrophil-like cells. Similarly, in lines 28-30 and lines 148-149 they cite a host of different papers that also model directional sensing in eukaryotic chemotaxis that they dismiss but should address. They state that these models cannot capture memory in polarization along with continuous adaptation or require fine-tuning. But it is not apparent that this is the case. The work by the authors is sufficiently different from many of these other papers and their contribution is significant that it merits dissemination (even if as a different framework for the same problem). But they should acknowledge clearly what their specific contribution is, or how it performs better than other models. If possible, they should cite experimental evidence that distinguishes their predictions from those of other models.

Detailed notes regarding figures and text:

The figures and the captions are very unclear, and the text can be bigger. The captions do not properly explain what is happening (please use complete sentences for captions). A few (but not exhaustive) notes:

Figure 1: Please label the color notation for the bars indicating the direction of the gradient (red, green, and orange) or indicate it somewhere in the caption. A plot similar to Figure 2B would be very helpful.

Figure 1A: This figure is highly confusing without the text. As such it served no purpose for me, and even after reading the text and understanding what it was trying to say, I didn’t understand the figure itself. The caption labels are inconsistent with the coloring in the plot, and there’s a notion of time that was not conveyed in the figure. It might serve the reader to split this into two figures and make the figure captions more explanatory.

1B: Not all the different elements are not explained in the caption (basically everything inside the cell is unlabelled).

1C: The circuit is too small to see.

Supp Figure 1-1: Please label the figures in the correct order (the panels labeled E should alphabetically appear after F and G). Also, it might be beneficial to keep the same scale for Ep (or switch to Ep/Et while identifying Et in the figure itself).

It was difficult to find the parameters used for the IHSS. I would recommend a table of parameters with columns for the different sets of simulations performed, with descriptions of each of the parameters/variables.

1-1B The current description gives the impression that Et is titratable (but as I understand it, the total protein concentration was conserved in the mass-action kinetics) especially as it is placed along with Ep which does change in the simulations. Please elaborate on the description in the figure caption and distinguish Ep and Et as a variable and parameter respectively.

Figure 2A: Please indicate a color bar for EGF just as was provided for EGFRp.

2C: During what time was the spatial projection generated?

D, E, H: Please spell out that n and N refer to the sample size and the number of replicates for the general readership.

2F: The figure is barely legible. Please magnify, and consider normalizing E-Ep.

2G: What are the units for the area? Is it a fraction of the total area?

2H: What does the horizontal line at the solidity of 0.65 represent? Also, how is the "end of memory" determined?

Figure 3A: Please identify the memory phase and label it in blue.

Figure 4D Please add a horizontal line for cos\theta = 0.

4E This figure was very difficult to read. Please consider breaking it into multiple 2D plots as one of the dimensions in the plot currently serves no purpose

Note for Section 1:

This section is fairly heavy on jargon and technical language. While I appreciate the transparency, it might benefit the general reader to reduce the number of terms used. For example, stable inhomogeneous state regime, stable polarization, and inhomogenous steady-state are used interchangeably in the text and the figures. Sticking to a small set of terminology would be beneficial to the reader.

Notes for Section 2:

I would define solidity early on (as of now, the first definition appears in line 557, but solidity appears as a quantity as early as Figure 2). In line 168, I would clarify if it is the polarisation that is shallow or the gradient of EGF. Also, a key piece of evidence that is currently buried is the timescale of gradient washout. It would serve the reader to highlight it in the text.

Please also clarify the relevance of the Takens delay-embedding trajectory. Currently, I found it misleading since it merely states that is a period of transition between two distinguishable states: the polarised state and the unpolarised state, which is already captured in the time plot of the EGFR dynamics.

Notes for Section 3:

It is currently stated that "directed migration persisted for a transient period of time after the gradient wash-out". This doesn't seem to be quantified, and what constitutes a "transient" period is unclear. Thus, the statement ,"The directionality estimated in the 9h time-frame after the gradient removal was greater than the one in continuous stimulus absence" seems to contradict the statement that "After the memory phase, the cells transitioned to a migration pattern equivalent to that in the absence of a stimulus" as the duration of the transient period of transition is unspecified. Also, this statement is not adequately reflected in the statistics of the directionality (Figure 3B).

Notes for Section 4:

This section was well-written, with the exception of describing what the KDE distributions reveal. This is a key piece of statistical evidence that must be more clearly shown and its relevance discussed.

*Reviewer #2 (Recommendations for the authors):*

– In general, the experiments conducted support the key features of the dynamical model. It would strengthen the authors' conclusions if the effects of perturbations (e.g., by Lapatinib) could be clarified in the main text within the context of the model. For instance, is only the memory lost, or is the cell's ability to polarize in the presence of a gradient also disrupted?

– It would be helpful if one did not have to refer to the caption to read several of the figure panels. (An example: the color-coding in 3A, D).

*Reviewer #3 (Recommendations for the authors):*

Timescale of memory

In my understanding, the lifetime of a 'ghost' state near a saddle-node bifurcation goes as 1/r^(1/2), where r is the distance to the critical point. This suggests that the timescale associated with the memory state is sensitive to how close the system is to the critical point, and, at least formally, diverges as the system actually approaches this point. This would seem to present a fine-tuning problem: not only does the system need to have parameters tuned to be near criticality, but in fact they have to be exactly the right distance away to achieve a physiologically reasonable intermediate memory timescale. It would be useful for the authors to discuss this: how is the memory timescale controlled? How sensitive is it to parameter changes? Is this somehow a feature, in that the cell can potentially physiologically change its memory timescale?

Suggestions regarding the theoretical exposition:

1. The authors should clarify the dimensional reduction that led to the bifurcation diagrams in Figure 1A and Figure 1—figure supplement 1B. Absent additional justification of this reduction, it would also be clearer to describe this as an approximate treatment of the system (whose behavior is born out by the reaction-diffusion simulations), rather than a proof (line 100).

2. It would be useful to readers to include a more concrete discussion of the EGFR model in the main text, including which features of it drive the behavior of the system and which biochemical parameters control location in the phase space. Additionally, how does the magnitude of the external gradient affect the cell? As a suggestion, the authors could consider moving Methods 5.15, Equation 17, to the main text, with a description of what the important variables are, and what features are important to the presence of the pitchfork bifurcation.

Suggestions regarding the presentation of the measurements:

1. Related to the public review, the authors could strengthen the paper by explicitly discussing the discrepancies between the measurements and the expectations from the model, and potential explanations. Does experimental noise interfere with the EGFR phosphorylation profile measurements? Do the authors believe that some of the cells are not at criticality?

2. The fact that the model depends on cells being biochemically poised near a critical point suggests a variety of stronger experimental tests of the framework. As one example, the authors' analysis suggests that overexpressing EGFR should push the cells away from the critical point, into the inhomogeneous steady-state regime, where they would break symmetry according to the first-encountered gradient and no longer be capable of adapting to a new gradient. This would be quite surprising, as it would correspond to breaking a sensing system by increasing the number of sensors. While performing this experiment is likely beyond the scope of this paper, the authors could strengthen the presentation by discussing this and/or other more counterintuitive predictions of their model in light of existing empirical data and/or future experiments.

3. Figure 2—figure supplement 1C: given that the direction of the polarization relative to the gradient is important, it would be interesting to see all the polarization profiles (and the variability from cell to cell with respect to the direction relative to the gradient).

---

## [Author Response]

Essential revisions:The reviewers broadly appreciate the theoretical framework here and the new experimental data gathered. However, a few critical changes are required before publication.1) The most critical change suggested by the reviewers is tempering and clarifying the relationship between experimental results and theory here. The theoretical work here is already sufficiently interesting, and the experimental results add to it, but the current statements imply a consistency between theory and experiments that is not fully supported.The authors should temper their claims and clearly delineate the ways in which the experimental results agree and disagree with the theoretical model here. (`All models are wrong, but some are useful.') Such a clear statement would allow future work to build on the results here.2) The authors should discuss other predictions of their model for future experiments (e.g., see reviewer #2's and #3's comments).3) Simplify/reduce technical jargon – e.g., see reviewer #1's comments.

We thank the referees and the editor for their insightful and detailed suggestions that we directly followed in order to generate the amended version of the manuscript. We now particularly discuss the similarities and possible differences between the theory and the experimental observations, more strongly highlight several proposed predictions, but also provide additional predictions of our model with respect to cellular sensing and responsiveness to simultaneous stimuli. We also aimed to reduce the technical jargon and in general, restructure the results as suggested by the referees, as well as provide additional conceptual schemes in order to present the proposed dynamical mechanism and the results in a clear manner to improve the readability of our manuscript. We next provide detailed response to the comments and concerns raised by the referees.

Reviewer #1 (Recommendations for the authors):Appraisal of results supporting Figure #1 and theoretical background:The theoretical background is sound and detailed. However, much of it rests on the choice of parameters as identified by the authors. It would be beneficial to see the dependence of the existence of saddle-node pitchfork bifurcations (and similarly the stable polarisation state, and homogeneous symmetric steady states) on key parameters (especially the total EGFR on the plasma membrane, and the total amount of the membrane-localized PTPRG and the ER-bound PTPN2).

Indeed, as the referee suggested, the two key parameters that determine the dynamical behavior of the system are the total EGFR and PTPRG concentrations on the plasma membrane of the cell, and we would like to include in this response also the response regarding the Figure 1—figure supplement 1C (bifurcation analysis of the EGFR network) that is raised later by the referee.

The dynamical behavior of the system is dependent on the ratio between EGFR and PTPRG concentrations at the plasma membrane, as PTPRG exhibits the main dephosphorylating activity of the receptor at the plasma membrane. In Figure 1—figure supplement 1B (Figure 1—figure supplement 1C in the amended version of the manuscript), we have chosen to show a bifurcation diagram with respect to the total EGFR concentration at the plasma membrane (EGFRt), since the structure of the bifurcation diagram remains the same whether one or the other parameter is varied. In order to substantiate this fact, we provide here a two-parameter bifurcation diagram of EGFR_t_ vs. PTPRG_t_ (Author response image 1) where we demonstrate that there is a large parameter range where pitchfork bifurcation (PB) can be observed in the system. Additionally, we also provide a two-parameter bifurcation diagram of EGFR_t_ vs. PTPN2_t_ (Author response image 1). This demonstrates that PB always exist above a threshold PTPN2_t_ concentration.

**Author response image 1. sa2fig1:** In-depth bifurcation analysis of the EGFR network. (**A**) Two-parameter bifurcation analysis (EGFRt vs. PTPRGt) characterizing the range where pitchfork bifurcations (black line) emerge. (**B**) Same as in a, only when PTPN2t instead of PTPRGt is considered.

As the referee notes, the total EGFR concentration is conserved in the system, and our hypothesis suggests that the EGFR_t_ concentrations in cells correspond to organization at criticality (before the SN_PB_, gray shaded region in Figure 1—figure supplement 1C in the amended version). In order to understand how the signal induced transition from stable unpolarized to stable polarized state emerges, we have used EGFR_t_ as a bifurcation parameter to reveal all possible dynamical regimes of the system.

For an EGFR_t_ concentration that correspond to the organization at criticality, the system displays a basal, unpolarized state, as it is organized in the homogenous steady state (HSS) regime (Figure1—figure supplement 1C). Upon presentation of a localized signal however, a characteristic unfolding of the Pitchfork bifurcation [1] occurs (Author response image 2). The HSS are pushed to the sides and the IHSS regime dominates (a general characteristic of IHSS solutions,[2]). Thus, for the same EGFR_t_ concentrations, the basal unpolarized state is now unstable and the system populates the IHSS branch – polarizes. Upon the signal removal, the system will return to the initial HSS regime, however in this case, transiting via the metastable “ghost” state, as schematically shown in Figure 1—figure supplement 1A in the amended version of the manuscript.

**Author response image 2. sa2fig2:** Generic principles of the symmetry breaking mechanism. Unfolding of the pitchfork bifurcation – the bifurcation diagram of the full system as in a., only when a local EGF gradient signal is present.

Due to the extent of this theoretical analysis and additional concepts necessary to explain the dynamical features of the system, we have chosen only to briefly discuss this in the methods (Section 5.16) and provide a schematic representation of the unfolding in the current version of Figure 1—figure supplement 1A, as well as a simplified version of the transitions in the current version of Figure1A. We strongly believe that including the discussion on these transitions in greater details will distract the reader from the main message of the paper. However, we are currently preparing an additional theoretical manuscript where we explain these details, as well as how it differs from the current models in the field (i.e. Turing, LEGI and Wave-pining) that will be available on bioRxiv in the upcoming month.

The model parameters used result from a detailed experimental characterization of the EGFR network from our previous, as well as experimental work of other groups [3,4,5], including the ligand binding/unbinding rates, recycling time of unphosphorylated receptors, the rates of autonomous/autocatalytic activation of the receptors, the specific phosphatases’ reactivity towards the receptor etc.

Also, the authors claim that in their model, "The cell polarity is sustained even when the EGF signal is briefly disrupted, but also, the cell is able to rapidly reverse the direction of polarization when the signal direction is inverted". It would be helpful if the authors indicated (either by a numerical plot, or analytical calculations if possible) the time scales of such responses in their model. Chiefly, it would help to know how long the signal would need to be disrupted before the polarity is no longer sustained, how long the signal direction needs to be inverted before the cell reverses polarity, and how long it takes for the cell to either reverse or lose polarity.

a) When the disruption of the signal is longer than the memory duration in single cells, the cells will not be able to integrate the information and establish long-term migration. In this case, a so called “stop-andgo” migration pattern will be established. In the amended version of the manuscript, we provide numerical simulations (Figure 4—figure supplement 1B and Figure 4 – video 2), as well as include corresponding discussion in the text.

b) In the case of reverse polarization, cell lose the polarity after the memory in polarization is lost (~20min on average after the removal of the third signal). The duration of the memory in the reversed polarization is quantified in Figure 4E.

c) We thank the referee for this suggestion. We have quantified the time necessary for cells to reverse polarity after presentation of the reversed signal. As shown in Author response image 3, from the KDE of the projection of the cell’s relative displacement angles during the presentation of the third, reversed signal (time point 365min), it can be seen that on average, majority of the cells reverse their direction in migration within 10 min after the reversal signal is presented (yellow, cos*θ* takes negative values). This information is now included in the manuscript.

**Author response image 3. sa2fig3:** Quantification of the time intevbal in which cells revert their direction of migration, as extracted from KDE distributions of reversal of gradient migration data in Figure 4 —figure supplement 1F.

Appraisal of results supporting conclusion #1:The authors claim that MCF7 cells maintain a memory of the direction of previously encountered signals through prolonged EGFR phosphorylation polarisation and temporal evolution of the cell protrusion that emerges from a saddle-node pitchfork bifurcation which maintains the system away from the steady states. The key pieces of evidence present claim that after subjecting the cells to a stable gradient of EGF for one hour, the cells remain polarised for ~40 minutes. While the experimental evidence shows that the cells are partially polarised for ~40 minutes after gradient washout, the polarisation is less than it is in the presence of the signal and steadily approaches the unpolarised state (this is most clearly seen in the single-cell trajectories supplied in Supp. Figure 2-1F). This is unlike the distinct persistence of polarisation shown in the numerical results (Figure 1F and G).

In order to address the referee’s comments on the evidence of existence of a SNPB “ghost”, we combine here the answer to this, with the answer to the next comment. The “ghost” state is manifested as transient trapping of the trajectory of the system (depicted through the EGFRp measurements) in the vicinity of the previously stable polarized steady-state (Figure 1F for numerical, and 2F for experimental results), before rapidly resetting to the basal state. In the amended version of the manuscript we provide several additional EGFRp time series profiles from single cell measurements (Figure 2—figure supplement 2 E, F). Note that the shape of the EGFRp profiles w/o Lapatinib treatment is similar to the one shown from the numerical simulation. The absolute levels of EGFRp in the presence of the signal (IHSS state) and after signal removal (memory state, SN_PB_ “ghost”) slightly differ, as also reflected in change of the polarized area after signal removal (Figure 2 —figure supplement 1G).

However, the EGFRp levels at the “ghost” state are transiently steadily maintained, before transiting to the basal unpolarized state. This is significantly different in the case when the kinase activity of the receptor is inhibited using Lapatinib and thereby the “ghost” state is lost (Figure 2G, average from single cells in Figure 2 —figure supplement 2B).

To provide a systematic analysis of the EGFR phosphorylation profiles and their features with and without Lapatinib washout (Figures 2 F, G), we fitted first single-cell EGFRp measurements in both cases with an inverse sigmoidal function ∫(x)=aoan+xn, and quantified the Hill coefficient *n*, as well as the time frame in which 50% of EGFR phosphorylation is lost (half-life) after EGF signal removal (Figure 2 —figure supplement 2E-G). Under normal conditions, ~30min were necessary to decrease EGFRp values by 50%, corroborating the presence of memory in EGFRp after signal removal, whereas the Hill coefficient was ~2.88, demonstrating that after the memory is lost, there is a rapid transition to the basal state (median R^2^ of the fits ~0.79). The same function (with a, a_o_, n chosen from the fit corresponding to the previous case) however could not describe the EGFRp profiles in the Lapatinib treatment experiment, as manifested by the poor goodness-of-fit values (median R^2^ ~ 0.33). This indicates that the shape of the EGFRp profiles in both cases is different. We therefore fitted the data in the Lapatinib case fixing only a to the values obtained by the control case, and leaving a_o_ and *n* as free parameters, as they determine the upper plateau and the steepness of the drop to the basal level. In this case, a Hill coefficient of n ~ 1.28 was identified from the fitting (median R^2^ ~ 0.84), whereas EGFRp levels dropped by 50% within 10min after EGF removal. These results clearly demonstrate that EGFRp is transiently maintained at steady levels after gradient removal, before rapidly transiting to the basal state (property of a “ghost” state), whereas in the Lapatinib case, the system gradually transits to the basal state, reflecting a continuous dephosphorylation process.

Lapatinib is an ATP-analog EGFR inhibitor that locks the tyrosine kinase domain into an inactive conformation, however it does not prevent EGF-induced dimerization of the receptor, as it has been demonstrated using a conformational EGFR sensor [6]. Thus, Lapatinib only affects the kinase activity and therefore, the EGFR phosphorylation profile after Lapatinib stimulation will be determined mainly by the catalytic activity of the phosphatases.

The catalytic activity of fully active PTPs is however two to three orders of magnitude higher than that of tyrosine kinases [18]. Thus, when the system is governed mainly by the phosphatase activity under Lapatinib treatment, EGFR phosphorylation will be reduced to 50% from the maximum level in ~10min. In contrast, the memory in EGFR phosphorylation polarization after removal of the EGF signal is maintained for ~40min on average (up to ~150min in single cells Figure 2—figure supplement 1G), indicating that the transient maintenance of EGFR phosphorylation cannot be explained with slow dephosphorylation.

To me, the experimental data suggest that the timescale of relaxation from the polarized state to the unpolarized state due to slow dephosphorylation kinetics is ~40 minutes. The authors should describe what the characteristic chemical timescales of the system are, and if persistence of ~40 minutes is uncharacteristically large. How do the experimental results reflect the existence of a metastable "ghost" state as in saddle-node bifurcations, rather than two steady states that take ~40 minutes to move between them as in a subcritical pitchfork bifurcation without a saddle-node? The authors attempt to answer this by using Lapatinib later on (which should be brought into this section), but the difference due to Lapatinib isn't established as data is shown only for one cell, and further, the unintended effects of Lapatinib on the cells hasn't been shown (a possible way to demonstrate this would be a control experiment with cells treated with Lapatinib for a short amount of time and then exposed to a gradient of EGF to show that there is a memory in such a case).

A) With respect to the first part of the comment, we would like to refer back to the answer on the previous comment. Moreover, the presence of the “ghost” state in the control and the respective absence in the Lapatinib treated cells is also reflected through the exemplary state-space trajectory reconstructions, where a three- vs. two-state transitions respectively are identified. We also thank the referee for the suggestion to include the Lapatinib characterization in Figure 2, which we followed in the amended version of the manuscript.

B) We thank the referee for the suggestion regarding the possible control experiment. However, as Lapatinib completely blocks the kinase activity of the receptor, pre-stimulation with Lapatinib renders the cells unresponsive to growth factor stimulation for ~96h [7] and therefore this experiment would not yield a possible control in our case. To address this point raised by the referee, we therefore resorted to quantification of the EGFRp temporal profile characteristics in the control and Lapatinib cases, as well as the time interval in which EGFRp is reduced by 50% in single cells after EGF removal, as discussed above.

Appraisal of results supporting conclusion #2:The authors claim that the transient memory encoded in the EGFR phosphorylation polarisation and cell protrusion translates to memory in directional migration up a linear EGF gradient that persisted after the gradient washout. The authors demonstrate that there is a transient memory in directional migration after subjecting the cells to 5 hours of a linear gradient (however, they do not initially identify the period of this memory in Figure 3A and 3B. They later suggest that it is 50 minutes after gradient washout); however, it is difficult to see if the memory encoded in the EGFR phosphorylation polarisation and cell protrusion is accurately reflected in the memory in directional migration due to the vast differences in the timescales of the experimental evidence for the two (the former is in a constant exposure for 1 hour, while the latter is for a constantly decreasing exposure for 5 hours).

We would like to note here that the memory in directional migration has been estimated from the cell migration traces in Figure 3A (Figure 3 —figure supplement 1D), by quantifying the projection of the cell’s relative displacement angle along the gradient direction (quantification in Figure 3C). Although it has not been technically possible to quantify EGFR phosphorylation polarization and cell migration simultaneously due to the necessity to use different magnification during imaging in both cases, we have addressed the issue whether the memory in EGFR phosphorylation polarization results in memory in directional migration by performing migration experiments under Lapatinib addition, as Lapatinib completely hinders EGFR kinase activity (Figure 3G-I). In addition to the provided directionality quantifications (Figure 3H), in the amended version of the manuscript we provided as well quantification of the projection of the relative displacement angles in the Lapatinib condition (Figure 3I, Figure 3 —figure supplement 2H), as well as additional quantification of memory in directional migration from single-cell trajectories in both cases (Figure 3 —figure supplement 3).

Moreover, we would like to note that regardless of the difference in the time for gradient stimulation, once the system reaches stable polarized state, the transitions that follow after the gradient removal to the basal state through the metastable state remain equivalent, as these are generic properties of the system.

Further, the evidence using Lapatinib is not conclusive, as even if we believe that there is a difference between the directionality with and without Lapatinib (Figure 3J), this does not establish the link between transient molecular states and transient directionality.

Lapatinib blocks the EGFR kinase activity without any effect on its structural properties [6], thus the difference in memory in directional migration after gradient wash-out in the case with and without Lapatinib establishes the link between memory in polarized EGFR signaling and memory in directional migration.

The evidence in Figure 3G is inconclusive as it is not sufficiently different from the case without Lapatinib and Takens's delay-embedding trajectory is based on just one cell.

We note that the averaged single-cell EGFRp profiles upon Lapatinib treatment are shown in the current Figure 2—figure supplement 2B. Together with the additional quantifications on the shape of EGFRp profile with and without Lapatinib, the results show that in the Lapatinib case, EGFR phosphorylation decreases after EGF removal in an exponential fashion. This corresponds to direct transition between the polarized and the basal state without intermediate phase-space trapping, as observed in the exemplary reconstructed state-space trajectory (now in Figure 2G).

In order to further verify that such transient trapping in phase space does not occur in the case when EGFR phosphorylation dynamics is guided mainly by the dephosphorylation of phosphates, such as under the Lapatinib treatment, we have performed additionally numerical simulations where we reduced the EGFR kinase activity through its autocatalysis rate after gradient removal, to mimic the effect of Lapatinib (*α*_2_ = 0.25). The results included in Figure 2 —figure supplement 2C, D as well as Figure 2 – video 3, show that equivalently as in the experiments, EGFRp gradually transits from the polarized to the basal state, without intermediate state-space trapping.

A key piece whose repercussions were not discussed much by the authors is that the gradient steepness was progressively decreased. The authors treated the entire period before the gradient washout as being the same, and I am still unclear as to what the effect of decreasing the gradient was.

One of the basic features for efficient polarization mechanism is the capability of sensing steep and shallow gradients with an offset. In order to demonstrate that the proposed mechanism fulfills these conditions, both in the numerical simulations (Figure 1D) as well as in the migration experiments (Figure 3,4), we established a dynamic gradient whose steepness changes over time. The results show that the SN_PB_ mechanism enables sensing gradients that dynamically change in time. This is a more complex form of a spatial signal as compared to the static gradient, for which cells also establish directional polarization and migration. We have used a static gradient as a second gradient in the experiments (and simulations) shown in Figure 4.

Appraisal of results supporting conclusion #3:The authors claim that MCF10A cells can sense dynamic gradients while maintaining robust directional migration even when the signal was disrupted. This is seen very clearly in silico in Figure 4B, but the experimental results in Figure 4C clearly differ in that the cells do not seem to have any significant memory following the gradient washout.

Although we fully agree with the referee that there is discrepancy between the simulations and experiments, we would like to underline here that the discrepancy refers to the migration distance between the signals, and not a lack of memory in the experimental case. In order to verify this, we have calculated the displacement of the cells in the intervals when the signal is disrupted from the experimental tracks in Figure 4 —figure supplement 1F and this ranges on average between 9-30μm, in some cells even 45μm. That cells migrate and maintain the directionality in these intervals can be also seen from the respective KDE distributions (Figure 4E). We have also included several magnified single cell traces in the amended version of Figure 4 —figure supplement 1G to support these statements.

Further, the ability of the cells to respond to changes in the environment should be contrasted to the case of slow dephosphorylation kinetics to highlight the role of the transient memory state. In general, it would be helpful to understand what the effect of a shorter time scale of the transition from a polarised to unpolarised state would be on the robustness of the system, and what the effect of a longer time scale would be on the adaptation of the system.

As already described, due to the inhibition effect of Lapatinib on the kinase activity of the receptor, the case when cells were subjected to Lapatinib during gradient wash-out describes the behavior of the system when guided mainly by dephosphorylation (absence of a “ghost” state). That in this case, the temporal EGFRp profile has a distinct shape, as well as there is no intermediate state-space trapping was verified both experimentally, as well as with numerical simulations (new Figure 2 —figure supplement 2C, D as well as Figure 2 – video 3). However, in order to test what will be the effect on cell migration if hypothetically dephosphorylation would be slower, we consider here additionally an exponentially decaying model where the dynamics of receptor phosphorylation R_p_ is given by Equation 1:

∂Rp∂t=l(x,t)α(Rt−Rp)−βRp+DRp∂Rp∂x2(1)

I(x,t) is the spatial-temporal signal input, *α* is an activation term, *β* – the dephosphorylation constant, D_Rp_ – the diffusion constant, and R_t_ – the total receptor amount. The value of *β* was chosen to capture a slow dephosphorylation profile after the signal removal (Author response image 4, left), in contrast to transition through a metastable state in the EGFR-PTP model (Figure 4A in the response, right). In order to demonstrate the difference this brings to spatial-temporal signal processing, two distinct sequences of gradient fields were used (Author response image 4), where the first two gradients (green) the same timing and duration, whereas the timing of the third gradient (red and blue) differs.

**Author response image 4. sa2fig4:** Optimal response to changing environments is rendered by metastable dynamics. (**A**) Response to single gradient stimulation using an exponential decay model for receptor activity dynamics (Equation 1) with slow dephosphorylation kinetics (left); and using the EGFR-PTP model with metastable “ghost” state. (**B**) Two distinct sequences of spatial-temporal gradient fields. (**C**) Directed protrusive area estimated from in-silico cell shape changes during stimulus sequence 1 (red) and sequence 2 (blue) for the exponential decay model (left) and EGFR-PTP model (right). Parameters for the exponential decay model: *α* = 1.5, R_t_ = 1, *β* = 0.5 × 10^−2^.

We performed viscoelastic simulations as described in Figure 1C in the manuscript, using the EGFR-PTP as well as the exponential decay model, to model the receptor activity dynamics, and quantified the morphological changes of the cell in both cases for both signals, using the directed protrusive area as in Figure 1G in the manuscript. When the receptor dynamics is guided by a slow dephosphorylation as in the exponential decay model, the cell morphology remains almost identical for the two distinct spatial-temporal signals (red and blue lines in Author response image 4 left correspond to morphological changes to signals 1 and 2 respectively), suggesting that the cell cannot distinguish between the two distinct signals. In contrast, when the receptor dynamics is guided by a “ghost” memory state in the EGFR-PTP model, the cell morphology directly follows the dynamics of the input signals (red and blue lines in Author response image 4 right correspond to morphological changes to signals 1 and 2 respectively), suggesting that this is a unique mechanism for cellular adaptation to varying spatial-temporal signals.

Note regarding the Discussion:The authors do not distinguish their contributions from those made by other research groups that are addressing similar questions. For example, in line 310, the authors cited a set of papers as relevant experimentally but dismissed them as not providing a mechanism even though some of them did. For instance, Skoge et al. (PNAS, 2014), discussed a possible mechanism by which cells exhibit memory and maintain directed motion in the classical back-of-the-wave problem for *Dictyostelium*, and Prentice-Mott et al. did so for chemotactic neutrophil-like cells.

We thank the referee for this suggestion. In the amended version of the manuscript, we have included a more detailed discussion how the mechanism of working memory proposed here differs with the work on the “back-of the-wave” problem in *Dictyostelium* and chemotactic behavior of neutrophil-like cells. In brief, in Skoge et al., 2014 [8], the authors propose mechanism that relies on LEGI model coupled to a memory-module (positive feedback leading to bistability). The duration of the memory is controlled (memory is turned on and off) through two independent thresholds. The authors however have not proposed a biochemical mechanism how the thresholds or their regulation can be realized, as they are crucial to enable a temporal memory in the polarized signaling activity. We have previously demonstrated that a bistability mechanism, when irreversible, will result in a permanent temporal memory. If the system displays reversible bistability, only memory of the input signal concentration can be exhibited, but not a temporal memory in the activity [9]. Moreover, in Skoge et al., 2014, potential molecular elements that store the information have also not been identified.

In the work of Prentice-Mott et al. [10], a novel mechanism of directional memory in migrating cells resulting from distinct molecular time scales has been proposed. The phenomenological model of transient memory in the cytoskeleton has however not been directly linked to a detailed biochemical mechanism. In the manuscript we present, we provide direct measurements of memory in receptor phosphorylation polarization which directs the memory in directional migration. Moreover, in the amended version of the manuscript, we included single-cell quantification of the duration of memory in directional migration in the case where the kinase activity has been blocked using Lapatinib, which effectively diminishes the memory in receptor phosphorylation. The obtained results are now discussed in relation to the results described in Prentice-Mott et al.

Similarly, in lines 28-30 and lines 148-149 they cite a host of different papers that also model directional sensing in eukaryotic chemotaxis that they dismiss but should address. They state that these models cannot capture memory in polarization along with continuous adaptation or require fine-tuning. But it is not apparent that this is the case. The work by the authors is sufficiently different from many of these other papers and their contribution is significant that it merits dissemination (even if as a different framework for the same problem). But they should acknowledge clearly what their specific contribution is, or how it performs better than other models. If possible, they should cite experimental evidence that distinguishes their predictions from those of other models.

We have decided to omit a detailed discussion on the differences between the SN_PB_ mechanism and previously proposed polarization mechanisms, as in our opinion, this requires an extensive analysis and a detailed discussion which would have significantly extended the scope of the manuscript. We have however performed such analysis by comparing several polarization features (i.e. capability for sensing steep and shallow gradients, threshold activation / spurious polarization to noise, signal amplification, time of polarization etc.) as well as information processing features (sensing subsequent signals, rapid repolarization, signal integration etc.) between the proposed SN_PB_ mechanism and several of the most common polarization models. i.e. Turing model [11, 12], LEGI [13] and Wave-pinning model [14]. The features used for the comparison were drawn from commonly observed features of migrating cells, i.e. leukocytes, neutrophils, Dictiostelium etc. as summarized in an extensive review by the Keshet lab [19]. The main difference between previously published, and the mechanism we propose here, is the organization of the system in parameter space: whereas all previous models are organized in stable attractor regimes, the SN_PB_ mechanism relies on organization at criticality and thereby utilizes metastable states for sensing and adaptation to changing signals. We present an example of these features in this response for the referee’s perusal, however, due to the extent of the analysis, we have chosen to summarize this in a related theoretical manuscript which will be posted on bioRxiv in the upcoming months.

As example we choose to present here the difference between the proposed SN_PB_ mechanism, LEGI, Turing and Wave-pinning models for adaptation to changing signals. First, we quantified the efficiency of the 4 models to respond to signal reversal (Author response image 5). Estimation of polarization amplification, which is the ratio of response amplitude at the leading edge of the cell before and after the reversal of gradient (RrevRfront) and the time required for reversing the polarization for each of the models demonstrated that rapid reversal is a feature that is unique for the SN_PB_ mechanism (Author response image 5). Neither the Wave-pinning model nor the Turing model were able to adapt the direction of polarization to the reversed gradient localization in physiologically relevant time, due to the permanent memory resulting from stable attractor organization. We also tested capability of the different models to integrate information from time-varying signals. Due to the same reason, Wave-pinning and Turing models show hindered responsiveness (response is equivalent whether one or two signals are presented, Author response image 5, left). The LEGI-model, due to the lack of memory, fails to integrate the information and responds to each input individually, and only the SN_PB_ model, due to the presence of metastable dynamics, shows capability to generate a response by integrating both signals (Author response image 5, right).

**Author response image 5. sa2fig5:** Comparison of information processing features between existing polarisation models (LEGI, Turing, Wavepinning) and the proposed SN_PB_ mechanism. (**A**) Schematic of gradient reversal. (**B**) Quantification of the time for polarization reversal and corresponding polarization amplification (RrevRfront) at stimulus ratio, (SrevSfront)=2. (**C**) Time series of signaling response at the leading edge (R_front1_) of the cell upon two consecutive gradients from the same direction for Wave-pinning and Turing models (left), and LEGI and SN_PB_ model (right).

Detailed notes regarding figures and text:The figures and the captions are very unclear, and the text can be bigger. The captions do not properly explain what is happening (please use complete sentences for captions).

We have now revised the figure legends, including all of the comments raised by the referee in order to improve the readability.

A few (but not exhaustive) notes:Figure 1: Please label the color notation for the bars indicating the direction of the gradient (red, green, and orange) or indicate it somewhere in the caption. A plot similar to Figure 2B would be very helpful.

We have included schematic representation for the gradient direction in the relevant figures.

Figure 1A: This figure is highly confusing without the text. As such it served no purpose for me, and even after reading the text and understanding what it was trying to say, I didn’t understand the figure itself. The caption labels are inconsistent with the coloring in the plot, and there’s a notion of time that was not conveyed in the figure. It might serve the reader to split this into two figures and make the figure captions more explanatory.

Following the referee’s suggestion, we have provided a more schematic representation of the figure in the amended version of the manuscript in order to explain the basis of the dynamical mechanism discussed, included a more detailed schematic representation in the current version of (Figure 1—figure supplement 1A), and improved the explanations in the figure captions.

1B: Not all the different elements are not explained in the caption (basically everything inside the cell is unlabelled).1C: The circuit is too small to see.

We have improved the size, added an additional legend and described in more details in the figure legend.

Supp Figure 1-1: Please label the figures in the correct order (the panels labeled E should alphabetically appear after F and G). Also, it might be beneficial to keep the same scale for Ep (or switch to Ep/Et while identifying Et in the figure itself).It was difficult to find the parameters used for the IHSS. I would recommend a table of parameters with columns for the different sets of simulations performed, with descriptions of each of the parameters/variables.

We have revised the labeling in the figure and also added a table of the parameters/ variables for the signaling, as well as the physical model of the cell.

1-1B The current description gives the impression that Et is titratable (but as I understand it, the total protein concentration was conserved in the mass-action kinetics) especially as it is placed along with Ep which does change in the simulations. Please elaborate on the description in the figure caption and distinguish Ep and Et as a variable and parameter respectively.

The answer to this question we have addressed in the answer to comment 1 by the referee and provided additional bifurcation analysis (Figure 1 in this response) where we clarify the raised issues.

Figure 2A: Please indicate a color bar for EGF just as was provided for EGFRp.

Amended.

2C: During what time was the spatial projection generated?

The spatial projection is calculated in the gradient duration period (5min – 65 min), as a temporal average per spatial bin. The basis of this calculation is also given in the methods section.

D, E, H: Please spell out that n and N refer to the sample size and the number of replicates for the general readership.

This is amended.

2F: The figure is barely legible. Please magnify, and consider normalizing E-Ep.

We have now enlarged the figure, however we option to keep the E-Ep profile normalized to the maximal fraction of ligand-bound receptors that can be achieved for the given EGF concentration – the position of the cell within the gradient, as estimated from the experimental data. As shortly also noted in the manuscript, the ligand bound fraction exponentially decays after EGF removal [3] in contrast to the EGFRp profile, demonstrating that the memory in EGFR phosphorylation is not due to residual EGF-bound receptor.

2G: What are the units for the area? Is it a fraction of the total area?

The directed cell protrusion area was estimated by comparing single cell masks at two consecutive time points. Protrusions were considered if the area change was greater than 10 pixels or 1.2µm^2^ per time point. The directed cell protrusion area was then obtained using Aprot,frontAfront−Aprot,backAback and thereby it is a fraction. The extended information is provided in the Methods section.

2H: What does the horizontal line at the solidity of 0.65 represent? Also, how is the "end of memory" determined?

The horizontal line is just a visual guide of the value of the solidity of cells before establishing the gradient. The memory duration in cell morphology was calculated from the single-cell solidity profiles, and corresponds to the time-point at which the solidity is below mean-s.d. estimated during gradient presence, and the relevant description is in Methods, section 5.11.

Figure 3A: Please identify the memory phase and label it in blue.

In the previous version of the manuscript, the duration of memory in directional migration was calculated from the KDE distribution of *cosθ*, that reflects the population distribution. In order to calculate the duration of memory in directed migration for single-cell tracks and color the tracks accordingly, in the amended version of the manuscript we have first smoothened the trajectories using the Kalman filter and identified the memory duration from the respective *cosθ* distribution for single cells (Methods, Section 5.12). The quantifications are given in Figure 3 —figure supplement 3, along with few color-coded trajectories. We have however chosen to keep the population averages in the main figure as before.

Figure 4D Please add a horizontal line for cos\theta = 0.

Amended.

4E This figure was very difficult to read. Please consider breaking it into multiple 2D plots as one of the dimensions in the plot currently serves no purpose

The unlabeled dimension in the figure denotes time and we have used the successive positioning of KDE distributions derived from the gradient migration experiment in different time intervals to show how they shift relative to the distribution from normal migration experiment which represent time = 0 min or pre-stimulation scenario. We apologize for failing to label this axis in the previous version of the manuscript, this is now amended in all KDE distribution plots, and the direction of time remains as previously, denoted above the distributions with an arrow.

Note for Section 1:This section is fairly heavy on jargon and technical language. While I appreciate the transparency, it might benefit the general reader to reduce the number of terms used. For example, stable inhomogeneous state regime, stable polarization, and inhomogenous steady-state are used interchangeably in the text and the figures. Sticking to a small set of terminology would be beneficial to the reader.

We thank the referee for his suggestions. In the revised version, we have tried to minimize the number of jargon specific words and also describe the results in a clearer manner to improve the readability.

Notes for Section 2:I would define solidity early on (as of now, the first definition appears in line 557, but solidity appears as a quantity as early as Figure 2). In line 168, I would clarify if it is the polarisation that is shallow or the gradient of EGF. Also, a key piece of evidence that is currently buried is the timescale of gradient washout. It would serve the reader to highlight it in the text.

We thank the referee for this suggestion. Solidity is now defined in the paper in relation to Figure 2, we have also specified in the text that the gradient of EGF is shallow (line 178), and added the information that the gradient is washed-out in 4-5min (line 169).

Please also clarify the relevance of the Takens delay-embedding trajectory. Currently, I found it misleading since it merely states that is a period of transition between two distinguishable states: the polarised state and the unpolarised state, which is already captured in the time plot of the EGFR dynamics.

A unique characteristic of the presence of a “ghost” state is a transient trapping of a phase space trajectory near the polarized steady state (Figure 1F, bottom). Reconstructing the phase-space trajectory from EGFRp traces in the control case indeed demonstrates a presence of a “ghost” state in phase space (Figure 2F, bottom), which is lost upon treatment with Lapatinib (Figure 2G, bottom). In order to further verify that such transient trapping in phase space does not occur in the case of a slow dephosphorylation, such as under Lapatinib treatment, as noted to one of the comments above, we also performed additionally numerical simulations where we reduced the EGFR kinase activity through its autocatalysis rate after gradient removal to mimic the effect of Lapatinib (*α*_2_ = 0.25). The results in Figure 2 —figure supplement 2C, D show that similarly to the Lapatinib experiments, the system directly transits from the polarized to the basal state without intermittent phase-space trapping.

Notes for Section 3:It is currently stated that "directed migration persisted for a transient period of time after the gradient wash-out". This doesn't seem to be quantified, and what constitutes a "transient" period is unclear.

In the amended version of the manuscript we tried to improve the readability of the text in general, and highlighted the fact that the transient persistence of the directional migration (memory) after gradient wash-out is quantified in Figure 3C and is ~50min. We have additionally quantified the duration of memory in directional migration from single cell cos *θ* profiles (Figure 3 —figure supplement 3).

Thus, the statement ,"The directionality estimated in the 9h time-frame after the gradient removal was greater than the one in continuous stimulus absence" seems to contradict the statement that "After the memory phase, the cells transitioned to a migration pattern equivalent to that in the absence of a stimulus" as the duration of the transient period of transition is unspecified. Also, this statement is not adequately reflected in the statistics of the directionality (Figure 3B).

We apologize if the wording has caused confusion, which we try to amend in the revised version. To clarify, that directionality calculated after gradient removal is significantly different (p<0.001) from the one calculated under continuous EGF absence suggests that cell after gradient removal do not immediately revert to the random walk migration as characteristic in the continuous gradient absence, but transiently persist with directional migration (memory), leading to the increased directionality values. In Figure 3C we have quantified that the memory in directional migration after gradient wash out is ~50min. Additionally, from the KDE estimations in Figure 3 —figure supplement 2C, it can also be seen that the KDE estimated after gradient removal approaches the one estimated in continuous absence only after the ~50min of memory phase (compare red and grey distributions, and blue and green).

Notes for Section 4:This section was well-written, with the exception of describing what the KDE distributions reveal. This is a key piece of statistical evidence that must be more clearly shown and its relevance discussed.

We have now included a more detailed description of Figure 4E in this section.

Reviewer #2 (Recommendations for the authors):– In general, the experiments conducted support the key features of the dynamical model. It would strengthen the authors' conclusions if the effects of perturbations (e.g., by Lapatinib) could be clarified in the main text within the context of the model. For instance, is only the memory lost, or is the cell's ability to polarize in the presence of a gradient also disrupted?

Corresponding to the referee’s suggestion, we have re-structured and better clarified, especially the results regarding the Lapatinib experiments. From the solidity quantification under Lapatinib condition (now in Figure 2I) and the included exemplary cell masks at distinct time-points, it is evident that the polarized cell shape is quickly lost after EGF removal and Lapatinib administration, in contrast to the normal case (Figure 2H). This suggests that loss of memory in EGFR phosphorylation polarization is reflected in loss of memory in polarized cell shape, and subsequently loss of memory in directional cell migration (Figure 3G), for which we also included additional quantification (Figure 3I, Figure 3 —figure supplement 2H). Moreover, as discussed above, we performed additional numerical simulations to mimic the effect of Lapatinib and show that in absence of memory, the system directly transits from the polarized to the basal state (Figure 2—figure supplement 2D), as experimentally identified in the reconstructed state-space trajectory.

– It would be helpful if one did not have to refer to the caption to read several of the figure panels. (An example: the color-coding in 3A, D).

We hope that the amended version has improved readability.

Reviewer #3 (Recommendations for the authors):Timescale of memoryIn my understanding, the lifetime of a 'ghost' state near a saddle-node bifurcation goes as 1/r^(1/2), where r is the distance to the critical point. This suggests that the timescale associated with the memory state is sensitive to how close the system is to the critical point, and, at least formally, diverges as the system actually approaches this point. This would seem to present a fine-tuning problem: not only does the system need to have parameters tuned to be near criticality, but in fact they have to be exactly the right distance away to achieve a physiologically reasonable intermediate memory timescale. It would be useful for the authors to discuss this: how is the memory timescale controlled? How sensitive is it to parameter changes? Is this somehow a feature, in that the cell can potentially physiologically change its memory timescale?

Indeed, plotting the EGFR^mCitine^ intensity at the plasma membrane vs. the total memory duration in single-cell EGFR^mCitrine^ phosphorylation polarization shows that higher the receptor duration, higher the memory length (Figure 6 in the response). Experimentally, we find a broad distribution of memory duration both, in polarized EGFR^mCitrine^ phosphorylation in MCF7-EGFR^mCitrine^ cells (Figure 2E), as well as memory in directional migration in MCF10A cells (Figure 3 —figure supplement 3A), suggesting that cell-to-cell variability in EGFR concentration generates a span of memory durations.

Due to the characteristics of the EGFR system – ligand-bound receptors are unidirectionally removed from the PM and undergo degradation, whereas ligandless receptor recycle, the total concentration of receptors on the PM will progressively decrease and thereby the system will move away from criticality. However, how long cells will maintain memory in directional migration and thereby organization at criticality depends on the duration and the concentration of the growth factor signal. We have extended the discussion on these points in the manuscript, and also suggest that it would be interesting to study whether receptor networks are self-organized at criticality, or the system evolved to this critical point as a means to have optimal information processing capabilities.

Suggestions regarding the theoretical exposition:1. The authors should clarify the dimensional reduction that led to the bifurcation diagrams in Figure 1A and Figure 1—figure supplement 1B. Absent additional justification of this reduction, it would also be clearer to describe this as an approximate treatment of the system (whose behavior is born out by the reaction-diffusion simulations), rather than a proof (line 100).

In the revised version of the manuscript, we have added the justification of the one-dimensional projection, as outlined in the response to the first comment by the referee above.

2. It would be useful to readers to include a more concrete discussion of the EGFR model in the main text, including which features of it drive the behavior of the system and which biochemical parameters control location in the phase space. Additionally, how does the magnitude of the external gradient affect the cell? As a suggestion, the authors could consider moving Methods 5.15, Equation 17, to the main text, with a description of what the important variables are, and what features are important to the presence of the pitchfork bifurcation.

We thank the referee for this valuable suggestion. However, due to the extent of the EGFR model/parameter descriptions, we have decided to maintain the equations in the Methods section, but we have extended the description of the model and the dynamical transitions crucial to explain the mechanism of cellular navigation we propose here.

Suggestions regarding the presentation of the measurements:1. Related to the public review, the authors could strengthen the paper by explicitly discussing the discrepancies between the measurements and the expectations from the model, and potential explanations. Does experimental noise interfere with the EGFR phosphorylation profile measurements? Do the authors believe that some of the cells are not at criticality?

In the amended version of the manuscript, we discuss in greater details the discrepancy between the polarization kinetics and amplitude of polarization in the experiments and simulations, the influence of biochemical and/or technical noise for phase space reconstruction, as well as the main differences between the migration trajectories in Figure 4. We underline however that the numerical simulations predict qualitatively the dynamics and behavior of the system.

2. The fact that the model depends on cells being biochemically poised near a critical point suggests a variety of stronger experimental tests of the framework. As one example, the authors' analysis suggests that overexpressing EGFR should push the cells away from the critical point, into the inhomogeneous steady-state regime, where they would break symmetry according to the first-encountered gradient and no longer be capable of adapting to a new gradient. This would be quite surprising, as it would correspond to breaking a sensing system by increasing the number of sensors. While performing this experiment is likely beyond the scope of this paper, the authors could strengthen the presentation by discussing this and/or other more counterintuitive predictions of their model in light of existing empirical data and/or future experiments.

As the referee suggested, for higher EGFR concentration on the membrane, the system will generate a permanent memory of the direction of the first encountered signal, which will render the cell unresponsive to further changes in signal localization (Figure 4 —figure supplement 1C,D and Figure 4 – video 3). In the amended version of the manuscript, we extended the discussion on this case, and we also provide an additional prediction – how cells resolve simultaneous signals. The numerical simulations predict that the dynamical memory that emerges from organization at criticality enables cells to compare and thereby resolve simultaneous signals with different amplitudes from different spatial localizations, whereas this feature is lost for organization in the IHSS regime (Figure 4 —figure supplement 2).

3. Figure 2—figure supplement 1C: given that the direction of the polarization relative to the gradient is important, it would be interesting to see all the polarization profiles (and the variability from cell to cell with respect to the direction relative to the gradient).

In the amended Figure 2 —figure supplement 1F, we provide a summary of the direction of polarization with respect to the gradient direction.

References:

1. Golubitsky, M., & Schaeffer, D. G. (1985). Singularities and Groups in Bifurcation Theory. 51. https://doi.org/10.1007/978-1-4612-5034-0

2. Koseska, A., Volkov, E., & Kurths, J. (2010). Parameter mismatches and oscillation death in coupled oscillators. Chaos: An Interdisciplinary Journal of Nonlinear Science, 20(2), 023132. https://doi.org/10.1063/1.3456937

3. Stanoev, A., Mhamane, A., Schuermann, K. C., Grecco, H. E., Stallaert, W., Baumdick, M., Brüggemann, Y., Joshi, M. S., Roda-Navarro, P., Fengler, S., Stockert, R., Roßmannek, L., Luig, J., Koseska, A., & Bastiaens, P. I. H. (2018). Interdependence between EGFR and Phosphatases Spatially Established by Vesicular Dynamics Generates a Growth Factor Sensing and Responding Network. Cell Systems, 7(3), 295-309.e11. https://doi.org/10.1016/J.CELS.2018.06.006

4. Reynolds, A. R., Tischer, C., Verveer, P. J., Rocks, O., & Bastiaens, P. I. H. (2003). EGFR activation coupled to inhibition of tyrosine phosphatases causes lateral signal propagation. Nature Cell Biology, 5(5), 447–453. https://doi.org/10.1038/NCB981

5. Baumdick, M., Brüggemann, Y., Schmick, M., Xouri, G., Sabet, O., Davis, L., Chin, J. W., & Bastiaens, P. I. H. (2015). EGF-dependent re-routing of vesicular recycling switches spontaneous phosphorylation suppression to EGFR signaling. eLife, 4. https://doi.org/10.7554/ELIFE.12223

6. Baumdick, M., Gelléri, M., Uttamapinant, C., Beránek, V., Chin, J. W., & Bastiaens, P. I. H. (2018). A conformational sensor based on genetic code expansion reveals an autocatalytic component in EGFR activation. Nature Communications 2018 9:1, 9(1), 1–13. https://doi.org/10.1038/s41467-018-06299-7

7. Wood, E. R., Truesdale, A. T., McDonald, O. B., Yuan, D., Hassell, A., Dickerson, S. H., Ellis, B., Pennisi, C., Horne, E., Lackey, K., Alligood, K. J., Rusnak, D. W., Gilmer, T. M., & Shewchuk, L. (2004). A unique structure for epidermal growth factor receptor bound to GW572016 (Lapatinib): relationships among protein conformation, inhibitor off-rate, and receptor activity in tumor cells. Cancer Research, 64(18), 6652–6659. https://doi.org/10.1158/0008-5472.CAN-04-1168

8. Skoge, M., Yue, H., Erickstad, M., Bae, A., Levine, H., Groisman, A., Loomis, W. F., & Rappel, W. J. (2014). Cellular memory in eukaryotic chemotaxis. Proceedings of the National Academy of Sciences of the United States of America, 111(40), 14448–14453. https://doi.org/10.1073/PNAS.1412197111

9. Stanoev, A., Nandan, A. P., & Koseska, A. (2020). Organization at criticality enables processing of time-varying signals by receptor networks. Molecular Systems Biology, 16(2), e8870. https://doi.org/10.15252/MSB.20198870

10. Prentice-Mott, H. v., Meroz, Y., Carlson, A., Levine, M. A., Davidson, M. W., Irimia, D., Charras, G. T., Mahadevan, L., & Shah, J. v. (2016). Directional memory arises from long-lived cytoskeletal asymmetries in polarized chemotactic cells. Proceedings of the National Academy of Sciences of the United States of America, 113(5), 1267–1272. https://doi.org/10.1073/PNAS.1513289113/SUPPL_FILE/PNAS.1513289113.SM15.MOV

11. Goryachev, A. B., & Pokhilko, A. v. (2008). Dynamics of Cdc42 network embodies a Turing-type mechanism of yeast cell polarity. FEBS Letters, 582(10), 1437–1443. https://doi.org/10.1016/J.FEBSLET.2008.03.029

12. Otsuji, M., Ishihara, S., Co, C., Kaibuchi, K., Mochizuki, A., & Kuroda, S. (2007). A Mass Conserved Reaction–Diffusion System Captures Properties of Cell Polarity. PLOS Computational Biology, 3(6), e108. https://doi.org/10.1371/JOURNAL.PCBI.0030108

13. Parent, C. A., & Devreotes, P. N. (1999). A cell’s sense of direction. Science, 284(5415), 765–769. https://doi.org/10.1126/SCIENCE.284.5415.765

14. Mori, Y., Jilkine, A., & Edelstein-Keshet, L. (2008). Wave-Pinning and Cell Polarity from a Bistable Reaction-Diffusion System. Biophysical Journal, 94(9), 3684. https://doi.org/10.1529/BIOPHYSJ.107.120824

15. Koseska, A., & Bastiaens, P. I. H. (2020). Processing Temporal Growth Factor Patterns by an Epidermal Growth Factor Receptor Network Dynamically Established in Space. Https://Doi.Org/10.1146/Annurev-Cellbio-013020-103810, 36, 359–383. https://doi.org/10.1146/ANNUREV-CELLBIO-013020-103810

16. Ibach J., et al.,(2015). Single particle tracking reveals that EGFR signaling activity is amplified in clathrin coated pits. PloS One 10, e0143162.

17. Stoscheck, C. M., & Carpenter, G. (1984). Characterization of the metabolic turnover of epidermal growth factor receptor protein in A-431 cells. *Journal of Cellular Physiology*, *120*(3), 296–302. https://doi.org/10.1002/JCP.1041200306

18. Fischer, E. H., Charbonneau, H., & Tonks, N. K. (1991). Protein tyrosine phosphatases: a diverse family of intracellular and transmembrane enzymes. *Science*, *253*(5018), 401–406. https://doi.org/10.1126/SCIENCE.1650499

19. Jilkine, A., Edelstein-Keshet, L., (2011). A comparison of mathematical models for polarization of single eukaryotic cells in response to guided cues. PloS Computational Biology. https://doi.org/10.1371/journal.pcbi.1001121